# Optimizing 4D Gaussians for Dynamic Scene Video from Single Landscape Images

**In-Hwan Jin**[1][*]   **Haesoo Choo**[2][*]   **Seong-Hun Jeong**[1]
**Heemoon Park**[3]   **Junghwan Kim**[4]   **Oh-joon Kwon**[5]   **Kyeongbo Kong**[1][†]
[1]Pusan National University   [2]Pukyong National University
[3]Busan Munhwa Broadcasting Corporation   [4]Korea University   [5]DM Studio

## Abstract

To achieve realistic immersion in landscape images, fluids such as water and clouds need to move within the image while revealing new scenes from various camera perspectives. Recently, a field called dynamic scene video has emerged, which combines single image animation with 3D photography. These methods use pseudo 3D space, implicitly represented with Layered Depth Images (LDIs). LDIs separate a single image into depth-based layers, which enables elements like water and clouds to move within the image while revealing new scenes from different camera perspectives. However, as landscapes typically consist of continuous elements, including fluids, the representation of a 3D space separates a landscape image into discrete layers, and it can lead to diminished depth perception and potential distortions depending on camera movement. Furthermore, due to its implicit modeling of 3D space, the output may be limited to videos in the 2D domain, potentially reducing their versatility. In this paper, we propose representing a complete 3D space for dynamic scene video by modeling explicit representations, specifically 4D Gaussians, from a single image. The framework is focused on optimizing 3D Gaussians by generating multi-view images from a single image and creating 3D motion to optimize 4D Gaussians. The most important part of proposed framework is consistent 3D motion estimation, which estimates common motion among multi-view images to bring the motion in 3D space closer to actual motions. As far as we know, this is the first attempt that considers animation while representing a complete 3D space from a single landscape image. Our model demonstrates the ability to provide realistic immersion in various landscape images through diverse experiments and metrics. Extensive experimental results are `https://cvsp-lab.github.io/ICLR2025_3D-MOM/`.

## 1 Introduction

When observing a real-world landscape, do you perceive the movement of the water and clouds? Can you discern how the speed of motion differs between distant mountains and nearby leaves as your perspective changes? If you notice these phenomena, it's because the actual landscape exists in a 4D space. Not only has significant research been conducted on creating realistic 3D videos by adding parallax to video based on depth information (Duong et al., 2019), but also to provide realistic immersion from static 2D images. Additionally, the synthesis of texture and structures in occluded regions can be enabled by 3D photography (Mildenhall et al., 2021; Wiles et al., 2020; Tucker & Snavely, 2020; Niklaus et al., 2019; Shih et al., 2020; Kopf et al., 2019), thereby allowing parallax effects from a single image. Among them, dynamic visual effects can be infused into static images by single-image animation (Chuang et al., 2005; Jhou & Cheng, 2015; Endo et al., 2019; Logacheva et al., 2020; Holynski et al., 2021; Fan et al., 2023) which enables fluid motion.

Recently, a field known as dynamic scene video (Li et al., 2023; Shen et al., 2023) has emerged, which creates videos with natural animations from specific camera perspectives using a combination of single image animation and 3D photography. These methods utilize Layered Depth Images (LDIs) (Shih et al., 2020; Kopf et al., 2019; 2020; Wang et al., 2022; Tulsiani et al., 2018), which are created by dividing a single image into multiple layers based on depth, to represent a pseudo

---

[*]Equal contribution.

[†]Corresponding author.

3D space. However, there are limitations when attempting to discretely separate most elements, including fluids, in a continuous landscape, and 3D space cannot be fully represented this way. Consequently, distortions can be observed, or the depth perception of the space can be diminished with camera movement. Therefore, the achievement of complete 4D space virtualization through explicit representation, rather than relying on LDIs, is necessary.

The recently emerged 3D Gaussian splatting (Kerbl et al., 2023) offers high-quality and efficient real-time rendering by representing scenes in 3D space through multiple 3D Gaussians using explicit representation. This research has also been expanded to 4D Gaussians (Luiten et al., 2023; Yang et al., 2024; Shaw et al., 2023; Huang et al., 2024; Liang et al., 2023; Ling et al., 2023; Yin et al., 2023; Ren et al., 2023), with a time axis added to 3D Gaussians to represent the structure and appearance of 3D objects over time. Among these research, some researchers (Wu et al., 2024; Yang et al., 2024; Huang et al., 2024) focused on modeling the changes in position, rotation, and scaling of each 3D Gaussian over time to achieve more natural motion in 4D Gaussians.

In this paper, we propose a method to represent a complete 3D space for dynamic scene video by modeling 4D Gaussians from a single image. As shown in Fig. 1, the proposed framework consists of three step: (1) 3D Gaussians Optimization, (2) Consistent 3D Motion Estimation, and (3) 4D Gaussian optimization. First, to optimize the 3D Gaussians, we generate multi-view RGB images from a single input image. At this stage, we use a 3D point cloud created by lifting the pixels of a 2D image into a 3D space. Subsequently, we aim to optimize 4D Gaussians by moving the regions corresponding to the motion areas of the optimized 3D Gaussians. To achieve this, we create a multi-view motion mask and then optimize 3D motion within 3D space. Finally, through the proposed 3D motion, we calculate the changes in position, rotation, and scaling of the Gaussians over time. As a result, we can optimize 4D Gaussians that maintains consistency across multiple views.

The most important part of our framework is optimization of 3D motion. Our goal is to move the 3D Gaussians, which requires motion within the 3D space. However, motion estimation directly within 3D space, such as with 3D point clouds or 3D Gaussians, is not only unexplored in existing research but also poses a highly challenging task due to the lack of dedicated datasets for this purpose. An alternative approach is to utilize existing off-the-shelf 2D motion estimation models (Holynski et al., 2021). Fortunately, motion estimation for 2D images has been extensively studied which allows the acquisition of motion from multi-view images to be easy. Therefore, we can easily achieve 3D motion by unprojecting the estimated 2D motion into the 3D domain using depth map.

However, when the estimated motion is applied to the 3D space, we found that the motion consistency across the multi-view images is not maintained. This inconsistency arises because the motion is estimated independently for each view. Consequently, artifacts in the 4D Gaussians can be caused by the movement of the Gaussians with the current 3D motion, potentially resulting in distortion in the rendered dynamic scene video. To address this issue, we propose a **3D Motion Optimization Module (3D-MOM)**. This model aims to optimize 3D motion to generate consistent motion across multi-view images. Specifically, arbitrary 3D motion is modeled and then optimized through the loss between the unprojected 3D motion and the estimated 2D motion.

Our module also benefits from utilizing 4D Gaussians to represent a fully 3D space with animation. Gaussian splatting offers a differentiable volumetric model that enhances depth perception, visual fidelity, and facilitates faster training times. Furthermore, the output in the form of Gaussian in the 3D domain can offer even greater versatility compared to the previous limitation to the 2D domain. From these advantages, our module creates realistic immersive dynamic scene video that ensure high visual quality for various landscape images.

In essence, our main contributions include:

- We propose a complete 3D space virtualization method for dynamic scene video, which is the first attempt to consider animation in 3D space. To achieve this, we optimize 4D Gaussians from a single landscape image to provide more immersive dynamic scene video.

- To animate 3D Gaussians, we estimate 2D motion from the multi-view images, and then unproject it back into the 3D domain to generate 3D motion. In this regard, we propose **3D-MOM** as a method to find a consistent motion among multi-view images.

- Our framework uses a Gaussian-based spatial representation to model a complete 3D space. This approach provides not only realistic depth perception through volumetric representation but also highly practical utility with explicit outputs.

## 2 RELATED WORK

### 2.1 NOVEL VIEW SYNTHESIS FROM SINGLE IMAGE

Novel view synthesis involves generating 3D scenes from existing images for new camera perspectives. Recent advances, particularly Neural Radiance Fields (NeRF) (Mildenhall et al., 2021), have enabled lifelike 3D scene generation from multi-view images. Research (Wiles et al., 2020; Weickert, 1999; Tucker & Snavely, 2020; Niklaus et al., 2019; Shih et al., 2020; Kopf et al., 2019; Wang et al., 2022) has also focused on synthesizing 3D scenes from single images by mapping them to point clouds and using inpainting for rendering new views. However, these methods face challenges in maintaining content consistency between frames, particularly for complex scenes. Layered depth images (LDIs) (Shade et al., 1998; Tulsiani et al., 2018; Shen et al., 2023; Li et al., 2023) have been proposed to efficiently represent 3D scenes by dividing images into depth-based layers, but these approaches often result in depth perception issues or artifacts. VividDream (Lee et al., 2024) extends this idea to 4D scene generation using stable video diffusion, though it lacks explicit 3D motion modeling. Therefore, continuous 3D virtualization across various angles is crucial. Our proposed method addresses these challenges, offering seamless novel view synthesis with full 3D space virtualization.

### 2.2 SINGLE IMAGE ANIMATION

Single image animation generates dynamic visual effects from static images, with a focus on animating landscapes containing fluid elements like clouds and water. Early research (Chuang et al., 2005) separated layers manually to create looping video textures, while later studies (Jhou & Cheng, 2015) used physical modeling to animate vapor-like objects. Recent advancements in deep learning have led to motion prediction within single images (Endo et al., 2019; Logacheva et al., 2020; Holynski et al., 2021), transforming them into animated video textures. Among these, Holynski et al. (Holynski et al., 2021) produced realistic looping animations using Eulerian flow fields, effectively approximating fluid motion. This approach continues to be expanded in subsequent research (Mahapatra & Kulkarni, 2022; Fan et al., 2023; Mahapatra et al., 2023; Li et al., 2023; Shen et al., 2023). Recent methods, such as Text2Cinemagraph (Mahapatra et al., 2023), also use Eulerian flow to simulate impressive fluid motion in single images. Additionally, 3D Cinemagraphy (Li et al., 2023) and Make-It-4D (Shen et al., 2023) employ joint learning of animation and novel view synthesis to generate natural fluid animations in 3D space. We apply Eulerian flow to fluids to achieve natural animations when generating 3D motion to optimize 4D Gaussians.

### 2.3 4D GAUSSIAN SPLATTING

The Gaussian Splatting introduced by Kerbl et al. (Kerbl et al., 2023) uses an explicit representation to address the performance issues of the implicit NeRF approach (Mildenhall et al., 2021). Research has expanded to include a time dimension on 3D Gaussians that capture dynamic changes over time. Luiten et al. (Luiten et al., 2023) implement frame-by-frame training to define position and rotation per timestep, which shows promising tracking results despite some consistency issues. Yang et al. (Yang et al., 2024) added a deformation field using canonical 3D Gaussians and MLPs, but faced longer training times. Wu et al. (Wu et al., 2024) attempted to reduce this by replacing multi-resolution hexplane and a lightweight MLP, though it struggled with accurate deformation predictions. Huang et al. (Huang et al., 2024) focused on sparse control points and represented the 4D scene through interpolation. Alternatively, Li et al. (Li et al., 2024) enhanced motion expressiveness by integrating temporal features into 3D Gaussians. Despite these advances, training generally relies on multi-view images. Our work, however, optimizes 4D Gaussians from a single image to generate and refine 3D motion.

## 3 PROPOSED METHOD

In this section, we provide an overview of our framework, as shown in Fig. 1. Our proposed pipeline consists of three main stages: 1) Generation of multi-view RGB images from a single image to optimize 3D Gaussians (Sec. 3.1). 2) Estimation of 3D motion in order to move the 3D Gaussians. Since the 2D motion maps estimated from the multi-view RGB images lack consistency in the

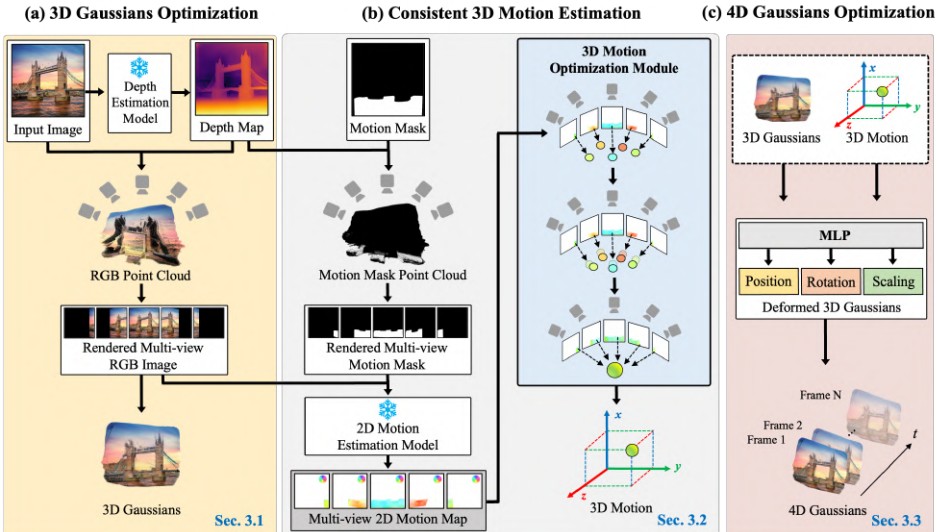

Figure 1: **The overview of our pipeline.** Our goal is to optimize 4D Gaussians to represent a complete 3D space, including animation, from a single image. (a) A depth map is estimated from the given single image, and it is converted into a point cloud. For optimizing the 3D Gaussians, multi-view RGB images are rendered according to the defined camera trajectory. (b) Similarly, multi-view motion masks are rendered for the input motion mask. These are utilized to estimate multi-view 2D motion maps along with the rendered RGB images. 3D motion is obtained by unprojecting the estimated 2D motion into the 3D domain. In this context, the proposed **3D Motion Optimization Module (3D-MOM)** ensures consistent 3D motion across multi-views. (c) Using the optimized 3D Gaussians and generated 3D motion, 4D Gaussians are optimized for changes in position, rotation, and scaling over time.

3D space, we propose a **3D Motion Optimization Module (3D-MOM)** to estimate consistent 3D motion (Sec. 3.2). 3) Warping the 3D Gaussians using the estimated 3D motion and optimizing the 4D Gaussians (Sec. 3.3).

## 3.1 MULTI-VIEW IMAGE GENERATION FOR 3D GAUSSIANS OPTIMIZATION

Recently introduced 3D Gaussian splatting (Kerbl et al., 2023) uses explicit representation to depict scenes in 3D space with numerous 3D Gaussians. This method, as a differentiable volumetric representation, provides depth perception from multi-views. We use 3D Gaussians to represent a complete 3D space from a single landscape image and optimize 4D Gaussians (Wu et al., 2024; Luiten et al., 2023; Yang et al., 2024; Shaw et al., 2023; Huang et al., 2024; Liang et al., 2023; Ling et al., 2023; Yin et al., 2023; Ren et al., 2023) by incorporating motion into the represented 3D space. To achieve this, we need to generate multi-view RGB images from a single image to optimize the 3D Gaussians.

### 3.1.1 POINT CLOUD GENERATION

To generate multi-view RGB images, it is necessary to convert 2D image into 3D space and project it according to various camera parameters. To achieve this, we first generate a point cloud from a single image. Specifically, we use a monocular depth estimation model to estimate the depth map $\mathbf{D} \in \mathbb{R}^{1 \times H \times W}$ for the input image $\mathbf{I} \in \mathbb{R}^{3 \times H \times W}$. Subsequently, we unproject the input image and the depth map into 3D space to create the point cloud $\mathcal{P}_I$:

$$\mathcal{P}_I = \{(\mathbf{X}_i, \mathbf{C}_i)\} = \Phi_{2 \to 3}(\mathbf{I}, \mathbf{D}; \mathbf{K}, \mathbf{E_0}), \tag{1}$$

where $\mathbf{X}_i$ and $\mathbf{C}_i$ are 3D coordinates and the RGB values for each point, respectively. $\Phi_{2 \to 3}(\cdot)$ is the function to lift pixels from the RGB image to the point cloud, and $\mathbf{K}$ and $\mathbf{E_0}$ are the camera intrinsic matrix and the extrinsic matrix of input image $\mathbf{I}$.

### 3.1.2 MULIT-VIEW IMAGE RENDERING AND 3D GAUSSIANS OPTIMIZATION

Inspired by prior works (Chung et al., 2023), we project the initial point cloud onto a 2D plane image $\tilde{\mathbf{I}}_i$ according to specific camera extrinsic parameters $\mathbf{E}_i$ to render multi-view RGB images as follows:

$$\tilde{\mathbf{I}}_i = \Phi_{3\to2}(\mathcal{P}_I, \mathbf{K}, \mathbf{E}_i). \tag{2}$$

Starting with the center camera view point $\mathbf{E_0}$, we continue the rendering process until all the cameras have been traversed. To handle holes introduced during rendering, we apply a simple linear interpolation, ensuring efficient and high-quality multi-view image generation. These generated multi-view images act as the ground truth (GT) for the multiview loss.

Using these rendered images, we initialize and optimize 3D Gaussians to represent the scene volumetrically. Following (Kerbl et al., 2023), we optimize the Gaussian parameters by balancing L1 and SSIM losses:

$$L = L1 + \lambda(L_{\text{SSIM}} - L1), \tag{3}$$

where $L_{\text{SSIM}}$ enhances structural similarity while $L1$ minimizes pixel-level discrepancies. This ensures accurate volumetric representation and prepares the Gaussians for subsequent motion and temporal modeling.

In Sec. 3.1, we have streamlined the discussion of 3D Gaussian optimization to focus on its role in bridging the gap between 3D scene generation (Chung et al., 2023; Ouyang et al., 2023) and our dynamic scene video. To enable the novel task of 4D scene generation, we designed Section 3.1 to closely align with prior 3D scene generation works. Further details on 4D scene generation will be provided in Sec. 4.5 Application: 4D Scene Generation.

## 3.2 CONSISTENT 3D MOTIONS ESTIMATION

To animate still 3D Gaussians, we need to estimate the corresponding motion field. However, direct 3D motion generation from a point cloud or 3D Gaussians is very challenging. Fortunately, various research has been conducted in the field of single-image animation (Chuang et al., 2005; Jhou & Cheng, 2015; Endo et al., 2019; Logacheva et al., 2020; Holynski et al., 2021; Mahapatra & Kulkarni, 2022; Fan et al., 2023; Mahapatra et al., 2023) which enable the estimation of 2D motion in areas where masks are provided. Therefore, we propose a novel approach that leverages existing off-the-shelf animation models to estimate 2D motion from multi-view images. Subsequently, we unproject this 2D motion into the 3D domain to estimate 3D motion.

### 3.2.1 MULTI-VIEW 2D MOTION ESTIMATION

Before estimating motion for multi-view images, it is necessary to generate a motion mask indicating the areas in each image where animation will be applied. Initially, we take an initial motion mask $\mathbf{M_0} \in \mathbb{R}^{1 \times H \times W}$ as input. Subsequently, similar to the existing equations (1) and (2), we generate a point cloud for the initial motion mask and project it to a specific camera view point $\mathbf{E}_i$ to create the motion mask $\tilde{\mathbf{M}}_i$ as follows:

$$\tilde{\mathbf{M}}_i = \Phi_{3\to2}(\Phi_{2\to3}(\mathbf{M_0}, \mathbf{D}; \mathbf{K}, \mathbf{E_0}); \mathbf{K}, \mathbf{E}_i). \tag{4}$$

To estimate motion maps from multi-view images at a subsequent time point, we adopt existing methods such as those by Holynski et al. (Holynski et al., 2021), which estimate the motion of fluids like water and clouds from a single image. Specifically, we use Eulerian flow $EF$ to represent the vector field that indicates the movement of pixels in the image, as follows:

$$\mathbf{F}_{t\to t+1}(\cdot) = EF(\cdot), \tag{5}$$

where $\mathbf{F}_{t\to t+1}(\cdot)$ represents the motion change from time $t$ to time $t+1$, and the Eulerian flow indicates that the amount of motion change between frames moves at a constant speed and direction over time. We estimate multi-view motion maps from generated multi-view images and motion masks using an existing 2D motion estimation model based on this Eulerian flow. This process can be formulated as follows:

$$\mathbf{F}_i = EF(\tilde{\mathbf{I}}_i, \tilde{\mathbf{M}}_i), \tag{6}$$

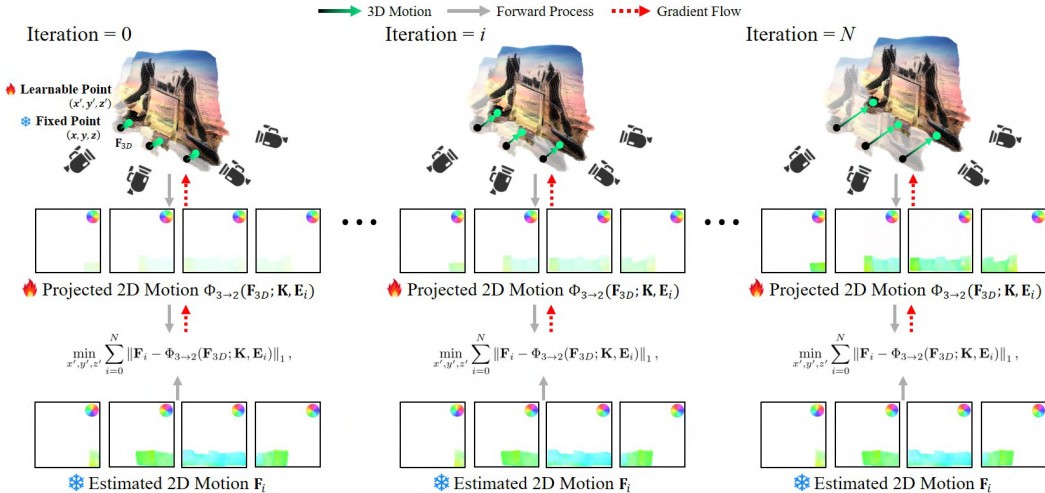

Figure 2: **3D Motion Optimization Module.** To maintain consistency of motion across multi-views, 3D motion is defined from the point cloud and projected into 2D images using camera parameters. The *L1 loss* between the projected 2D motion and the estimated 2D motion map as the ground truth is computed, minimizing the sum of losses for multi-view to optimize the 3D motion.

where $\tilde{\mathbf{I}}_i$ represents a multi-view image. Therefore, $\mathbf{F}_i$ denotes the motion map of the multi-view image $\tilde{\mathbf{I}}_i$.

However, since the multi-view motion maps $\mathbf{F}_i$ are estimated independently for each multi-view image, there may be a phenomenon known as ***Motion Ambiguity***. This occurs when the motion values at the same position in 3D space do not match when unprojected. Using these inconsistent motion maps directly for 4D reconstruction would result in artifacts and unnatural motion in dynamic regions. To address this, we propose the 3D-MOM, which leverages multi-view 2D motion maps to estimate consistent 3D motion, thereby extending traditional 2D motion methods into the 3D domain.

### 3.2.2 3D MOTION OPTIMIZATION MODULE

We prioritize achieving 3D motion consistency by parametrically modeling 3D motion. The modeled 3D motion is reprojected into 2D space, and its error is calculated relative to the estimated multi-view 2D motion maps. By optimizing this error while applying consistency constraints, our approach ensures that the resulting 3D motion is both consistent and robust, even though it may be sub-optimal in general 3D motion estimation tasks.

As illustrated in Fig. 2, we use the point cloud $\mathbf{P}$, located at coordinates $(x, y, z)$ as the starting point. Then, we define the coordinate difference between $(x', y', z')$ and $(x, y, z)$ as the 3D motion, $\mathbf{F_{3D}}$. This represents the movement, or motion, of the coordinate points in 3D space. Afterwards, we project the 3D motion information through camera parameters $\mathbf{K}$ and $\mathbf{E}_i$ into 2D image space, rendering it as $(u_i, v_i)$ in the 2D space. Subsequently, the L1 loss is computed between this projected motion $(u_i, v_i)$ and the motion map $\mathbf{F}_i$ derived from the 2D motion estimation model. Using the multi-view 2D motion $\mathbf{F}_i$ as the ground truth, we compute the loss in 2D space and optimize the 3D coordinate $(x', y', z')$ accordingly. This process can be formulated as follows:

$$\min_{x',y',z'} \sum_{i=0}^{N} \|\mathbf{F}_i - \Phi_{3\to2}(\mathbf{F}_{3D}; \mathbf{K}, \mathbf{E}_i)\|_1 , \tag{7}$$

where

$$\mathbf{F}_{3D} = (x', y', z') - (x, y, z) \tag{8}$$

$N$ represents the total number of multi-view images generated along the camera trajectory. To minimize (7), we updated $\mathbf{F_{3D}}$ through Stochastic Gradient Descent, ultimately obtaining consistent 3D motion in space.

### 3.3 4D GAUSSIANS OPTIMIZATION

Recently, the research in 3D Gaussian is expanded to 4D Gaussians (Luiten et al., 2023; Yang et al., 2024; Wu et al., 2024; Shaw et al., 2023; Huang et al., 2024; Liang et al., 2023; Ling et al., 2023; Yin et al., 2023; Ren et al., 2023) to represent the structure and appearance of 3D over time changes.

In our framework, we adopt 4D-GS (Wu et al., 2024) to ensure efficient training time and quality. However, our unified framework is not constrained to a specific 4D Gaussian model and is compatible with various models. As the field advances, it enables the rapid generation of high-fidelity dynamic scene videos. To predict the deformation of 3D Gaussians specifically, 4D-GS (Wu et al., 2024) utilize a Spatial-Temporal Structure Encoder. This enables the effective modeling of 3D Gaussian features using a multi-resolution HexPlane and a tiny MLP. Subsequently, through a multi-head Gaussian deformation decoder, we calculate the deformation of position, rotation, and scaling ($\Delta\mathbf{x}$, $\Delta\mathbf{r}$, $\Delta\mathbf{s}$), representing each as follows:

$$(\mathbf{x}', \mathbf{r}', \mathbf{s}') = (\mathbf{x} + \Delta\mathbf{x}, \mathbf{r} + \Delta\mathbf{r}, \mathbf{s} + \Delta\mathbf{s}), \tag{9}$$

where $\mathbf{x}, \mathbf{r}, \mathbf{s}$ are initial position, rotation, and scale, respectively.

#### 3.3.1 3D MOTION INITIALIZATION

In this case, to represent the animation, the most dominant component is the deformation of the position. This corresponds to the change in the mean values of each Gaussian, which is initialization with the previously estimated 3D motion. Therefore, by adding the motion information to the position in advance, we can modify the equation as follows:

$$(\mathbf{x}', \mathbf{r}', \mathbf{s}') = (\mathbf{x} + \mathbf{F_{3D}} + \Delta\mathbf{x}', \mathbf{r} + \Delta\mathbf{r}, \mathbf{s} + \Delta\mathbf{s}). \tag{10}$$

The remaining deformations, aimed at preventing distortion of the estimated 4D Gaussians, can also be sufficiently learned through weak supervision. Therefore, we jointly train the first frame's multi-view images $\tilde{\mathbf{I}}$ obtained earlier and the 2D animation results of the input image to estimate the remaining deformations. For this purpose, we draw inspiration from prior research (Mahapatra et al., 2023) (Fan et al., 2023) and utilize a video generation model to obtain the animation. In particular, they utilize the Eulerian flow field, which represents changes in motion over time moving at a constant speed and direction, to create realistic animated looping videos for fluids, as demonstrated by Holynski et al. (Holynski et al., 2021).

#### 3.3.2 TWO-STAGE TRAINING

Animating videos of the entire multi-view images is time-consuming and redundant due to significant overlap in areas. To address this, we devised a learning method that can accurately captures motion across the entire spatial and temporal domains while reducing time. Inspired by the training of prior work (Li et al., 2023; Shen et al., 2023), we applied a two-stage training approach across all viewpoints and animated videos. First, we fixed the time at 0 and trained a 3D Gaussian for the entire viewpoints. In the second stage, we trained 4D Gaussians across the entire temporal axis using video frames of sampled viewpoints. To the best of our knowledge, this is the first work to validate the effectiveness of a two-stage training approach for 4D Gaussian learning, demonstrating its capability to efficiently reconstruct spatiotemporal dynamics. Through this process, we can efficiently obtain 4D Gaussians that consider motion by applying natural animation for fluids from optimized 3D Gaussians. The experimental results for metrics and runtime can be found in Table 2 of Appendix.

## 4 EXPERIMENTS

### 4.1 EXPERIMENTAL SETUP

#### 4.1.1 BASELINE MODEL

To evaluate the effectiveness of our approach in the context of dynamic scene video, we compared it with two state-of-the-art models. 3D-Cinemagraphy (Li et al., 2023) utilizes LDIs for 3D representation and jointly produces animations to apply the parallax effect in realistic dynamic scene

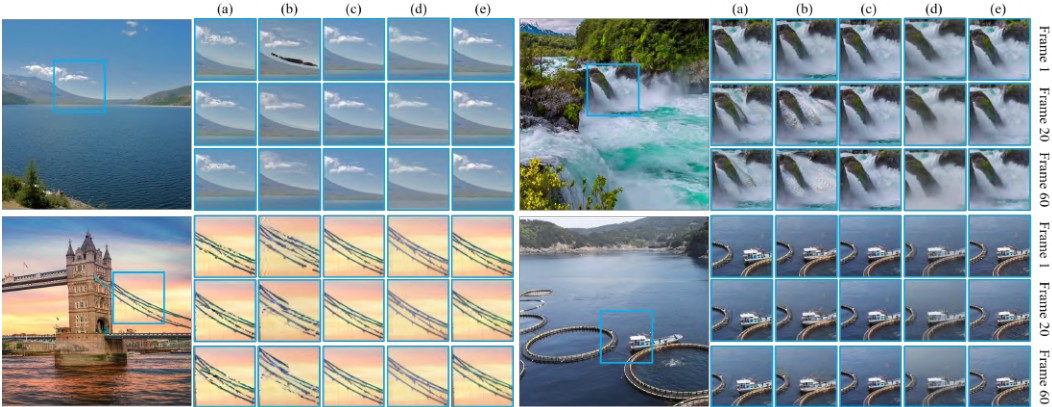

Figure 3: **Qualitative Results.** (a) 3D Cinemagraphy (Li et al., 2023), (b) Make-It-4D (Shen et al., 2023), (c) DynamiCrafter (Xing et al., 2025), (d) Motion-I2V (Shi et al., 2024), and (e) Ours.

Table 1: **Quantitative Results**. We measured three metrics to quantitatively compare the results of the baseline model and the proposed method. The results showed that the proposed model outperformed in all metrics. Furthermore, the user study evaluated the performance of each model, with the proposed model demonstrating the highest level of performance across all criteria.

| Method | Metrics | | | | User study (%) | | | |
|---|---|---|---|---|---|---|---|---|
| | PSNR ↑ | SSIM ↑ | LPIPS ↓ | PIQE ↓ | Immersion | Realism | Stuctural Consistency | Quality |
| DynamiCrafter (Xing et al., 2025) | 14.98 | 0.81 | 0.23 | 24.58 | - | - | - | - |
| Motion-I2V (Shi et al., 2024) | 14.38 | 0.80 | 0.31 | 8.40 | - | - | - | - |
| 3D-Cinemagraphy (Li et al., 2023) | 17.30 | 0.83 | 0.17 | 8.93 | 31.87 | 31.87 | 28.75 | 30.31 |
| Make-It-4D (Shen et al., 2023) | 16.98 | 0.81 | 0.20 | 8.30 | 11.87 | 10.31 | 8.43 | 9.06 |
| Ours | **20.57** | **0.90** | **0.14** | **7.80** | **56.25** | **57.81** | **62.81** | **60.25** |

video. Similarly, Make-It-4D (Shen et al., 2023) uses LDIs for 3D representation but it employs a pre-trained diffusion model for inpainting to fill occluded areas, achieving wider camera poses similar to long-range views. In these studies, the output is limited to the 2D domain in the form of images. In contrast, our model outputs in the 3D domain as Gaussian forms, which are projected to 2D according to camera trajectories for comparison of results.

### 4.1.2 IMPLEMENTATION DETAILS

We manually generate the input masks using the LabelMe tool to designate the motion areas in the input image. During multi-view image generation, we employ ZoeDepth (Bhat et al., 2023) for depth estimation, which provides normalized depth values. While these values lack absolute scale, they are sufficient for constructing a 3D point cloud when combined with intrinsic and extrinsic camera parameters. Intrinsic parameters are determined based on the input image size using standard conventions, and extrinsic parameters are initialized as an Identity Matrix to define the camera trajectory. We establish a rendering trajectory with about 30 camera viewpoints for point cloud rendering. To estimate flow from a single image, we utilize Holynski et al. (Holynski et al., 2021) and the pretrained single-image animation model from SLR-SFS (Fan et al., 2023), selected after evaluating various options for their ability to produce consistent looping animations and temporal coherence. Our 3D motion optimization module is trained for about 200 iterations with a batch size of 30 using the SGD Optimizer. We set the initial learning rate at 0.5 and then decayed it exponentially. All models were tested to ensure compatibility with the proposed framework. We conduct all experiments on a single NVIDIA GeForce RTX 3090 GPU.

### 4.1.3 EVALUATION DATASET

Following (Li et al., 2023), we evaluated our method and the baselines using the validation set from Holynski et al. (Holynski et al., 2021). The validation set consists of 162 samples of ground truth video clips, captured from a static camera viewpoint across 27 different scenes. For evaluation, we

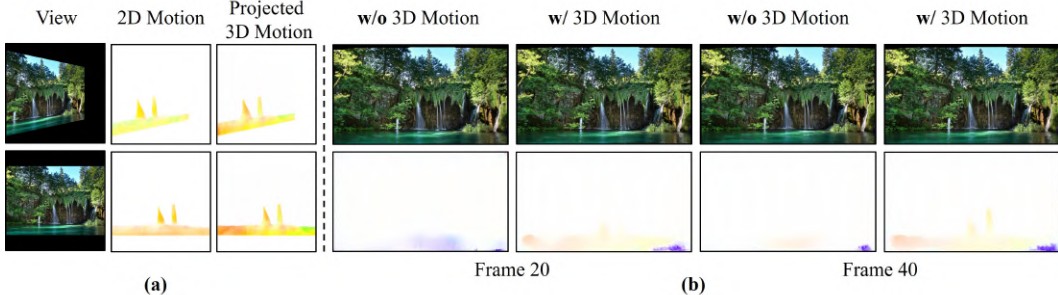

Figure 4: (a) Visual comparison of the 2D motions estimated from multi-view images and projected 3D motion estimated via 3D-MOM. (b) Rendered image of 4D Gaussians trained using viewpoint videos generated from 2D motion and 3D motion, and the optical flow between the resulting frames.

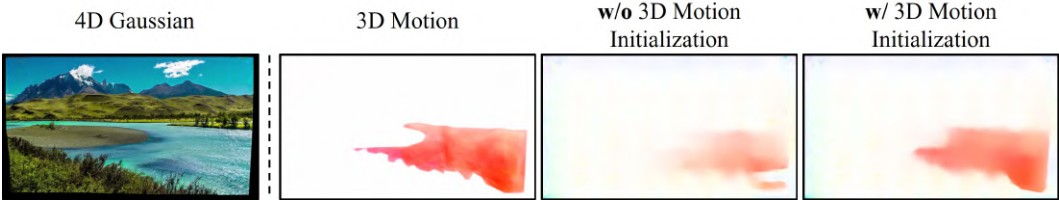

Figure 5: The effect of 3D motion initialization is applied to 4D Gaussians training. The left part shows the trained result frames and the projected 3D motion. The right part compares the results with and without 3D motion initialization by extracting the optical flow from the 4D Gaussians results at the same time and viewpoints.

rendered the ground truth videos from novel viewpoints using 4 different camera trajectories from 3D-Cinemagraphy (Li et al., 2023), resulting in 240 ground truth frames for each sample.

#### 4.1.4 EVALUATION METRICS

The generated videos are compared with the ground truth frames at the same time and from the same view. We use Peak Signal-to-Noise Ratio (PSNR), Structural Similarity Index Measure (SSIM) (Wang et al., 2004), and Perceptual Similarity (LPIPS) (Zhang et al., 2018) as reference metrics, and Perception-based Image Quality Evaluator (PIQE) (Venkatanath et al., 2015) as the non-reference evaluation metric for our study. We introduced a mask for the motion area to separately measure the moving regions. Additionally, in the ablation study, we utilized End-Point-Error (EPE), which is the average L2 distance between flows and is commonly used for measuring optical flow, to evaluate the effect of our 3D motion.

### 4.2 QUANTITATIVE RESULTS

In Table 1, we show the quantitative results of our method compared to other baselines on reference and non-reference metrics. Our approach outperforms the other baseline on all metrics in the context of view generation. In particular, our method achieved the highest scores in PSNR, SSIM, and LPIPS, indicating that the generated views are of high fidelity and perceptually similar to the ground truth views. Furthermore, we demonstrated that our proposed method outperforms existing methods on non-reference metrics by locally measuring the extent of noise and distortion in images using PIQE (Venkatanath et al., 2015). Additionally, we conduct a user study on the generated videos, confirming that our model not only outperform in quantitative metrics but also surpasses in user experiences across four visual aspects.

### 4.3 QUALITATIVE RESULTS

In Fig. 3, we present qualitative comparison results with other baseline methods and diffusion-based methods. In this case, our proposed model, as an explicit representation, is projected to 2D video for comparison of results. The process of separating the input image into LDIs in 3D-Cinemagraphy

(Li et al., 2023) leads to artifacts on animated regions and fails to provide natural motion which results in reduced realism. Similarly, Make-It-4D (Shen et al., 2023) also utilizes LDIs to represent 3D for multi-view generation, which results in lower visual quality. Additionally, due to unclear layer separation, objects appear fragmented or exhibit ghosting effects, where objects seem to leave behind afterimages. Likewise, DynamiCrafter (Xing et al., 2025) and Motion-I2V (Shi et al., 2024), though capable of producing cinemagraphy, encounter challenges in accurately rendering the desired views due to limited capabilities in view manipulation. In contrast, the proposed model represents a complete 3D space with animations, providing less visual artifact and high rendering quality from various camera viewpoints. Therefore, our method provides more photorealistic results compared to others for various input images.

## 4.4 ABLATION STUDY

**We encourage readers to visit the project page to explore our comprehensive video results.**

### 4.4.1 3D MOTION OPTIMIZATION MODULE

Independently estimated 2D motion from multi-view images can result in different motion values for the same region in 3D space. Directly using these 2D motions to animate viewpoint videos can fail to train 4D Gaussians to represent natural motion. The estimated motions are projected to the center point through a depth map for the same position measurement. Without the 3D Motion Optimization Module, the estimated flows show significant differences for the same positions, with an EPE of 0.152. On the other hand, with the 3D Motion Optimization Module, the consistency across entire viewpoints is outstanding, demonstrating an almost negligible variance with an EPE of 0.003.

Fig. 4 (a) shows the visualized results of 2D motion and projected 3D motion. Similarly, it indicates that 3D motion represents motion information in 3D space, which ensures consistency when projected to different viewpoints. We animated viewpoint videos using each motion and trained 4D Gaussians on multi-view videos. However, as shown in Fig. 4 (b), the rendered video of 4D Gaussians and estimated optical flow have the lack of motion consistency in the viewpoint videos caused unnatural movements.

### 4.4.2 EFFECT OF 3D MOTION INITIALIZATION

To verify the significance of 3D motion in training 4D Gaussians, we compared the results with and without 3D motion initialization. When applying animation to fluids, we observed repeated patterns. Fig. 5 shows rendered 3D motion and estimated optical flow from rendered video of each 4D Gaussian model. These results demonstrate challenges of accurately learning natural motion from videos without 3D motion initialization. When applying weakly supervised learning to deformation field by 3D motion, it accurately represents the motion from the generated multi-view videos.

## 4.5 APPLICATION: 4D SCENE GENERATION

To demonstrate the scalability of our framework, we extended its application to 4D scene generation by integrating it with existing 3D Scene Generation algorithms, such as ViewCrafter (Yu et al., 2024) and LucidDreamer (Chung et al., 2023). Experimental results, detailed in Appendix B, validate the effectiveness of our method in generating consistent and high-fidelity 4D scenes.

## 5 CONCLUSION

This paper proposes a method to model 4D Gaussians from a single landscape image to represent fluid-like motion within 3D space. Our approach efficiently estimates motion from multiple views using a 2D motion estimation model, and 3D-MOM optimizes the loss between 3D and 2D motion to produce consistent motion across views. The framework is applicable to various images, including those with complex depth variations, and its effectiveness has been validated through extensive experiments.

## 6 REPRODUCIBILITY

The code used in our research is available in full on the GitHub page at `https://github.com/cvsp-lab/ICLR2025_3D-MOM`, where detailed usage instructions are also provided. Masks and labeling for motion areas can be generated using the labelme program. The Holinsky dataset (Holynski et al., 2021), which was utilized in the research, is an existing public dataset that allows for experimentation on quantitative results. Furthermore, the data collected for qualitative results and further analysis have also been made publicly available. As a result, the findings of this paper are fully reproducible, and additional video results are uploaded at the provided link.

## 7 ACKNOWLEDGEMENTS

This work was supported by the National Research Foundation of Korea (NRF) grant funded by the Korea government (MSIT) (No. RS-2024-00456152) and regional broadcasting development support project funded by the Foundation for Broadcast Culture.

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

APPENDIX

## A    ADDITIONAL RESULTS ON "IN-THE-WILD" DATASET AND BASELINES

We verified our results qualitatively and quantitatively using Holinsky et al. (Holynski et al., 2021), which is commonly used for validation in the field of Dynamic Scene Video. Additionally, to compare the performance of our method with baseline models, we use our "in-the-wild" dataset, which we collect of global landmarks from online sources. Fig. 6,7 demonstrate that our model outperforms baseline models by producing more realistic and stable videos across a variety of complex scenarios.

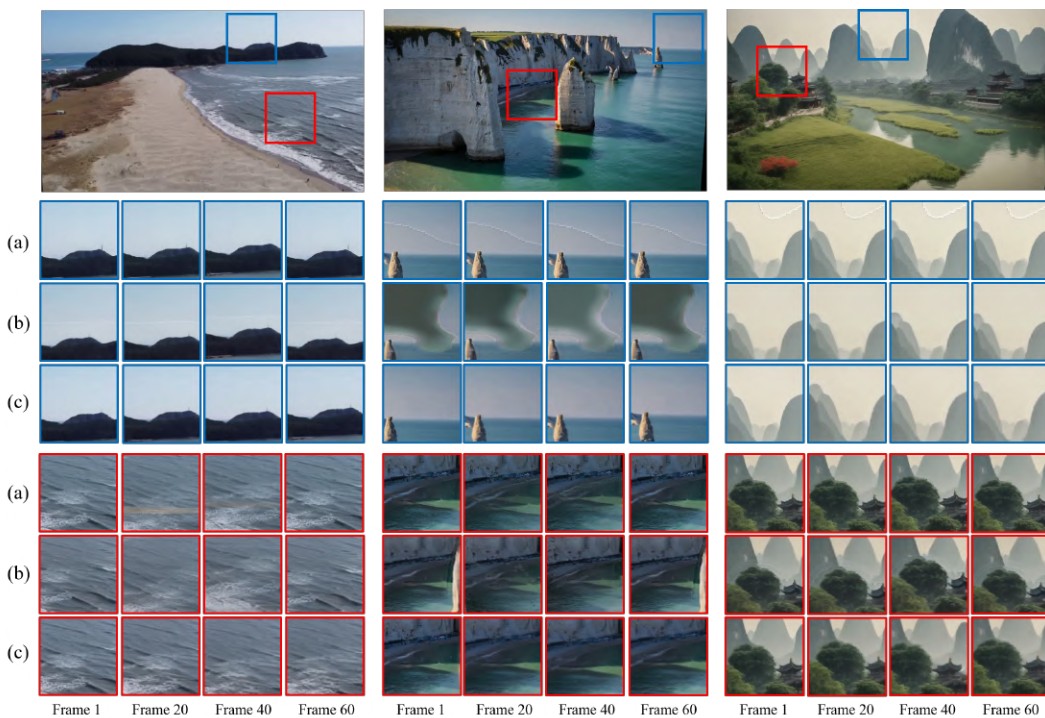

Figure 6: **"in-the-wild" dataset Results.** (a) 3D Cinemagraphy (Li et al., 2023), (b) Make-It-4D (Shen et al., 2023), (c) Ours

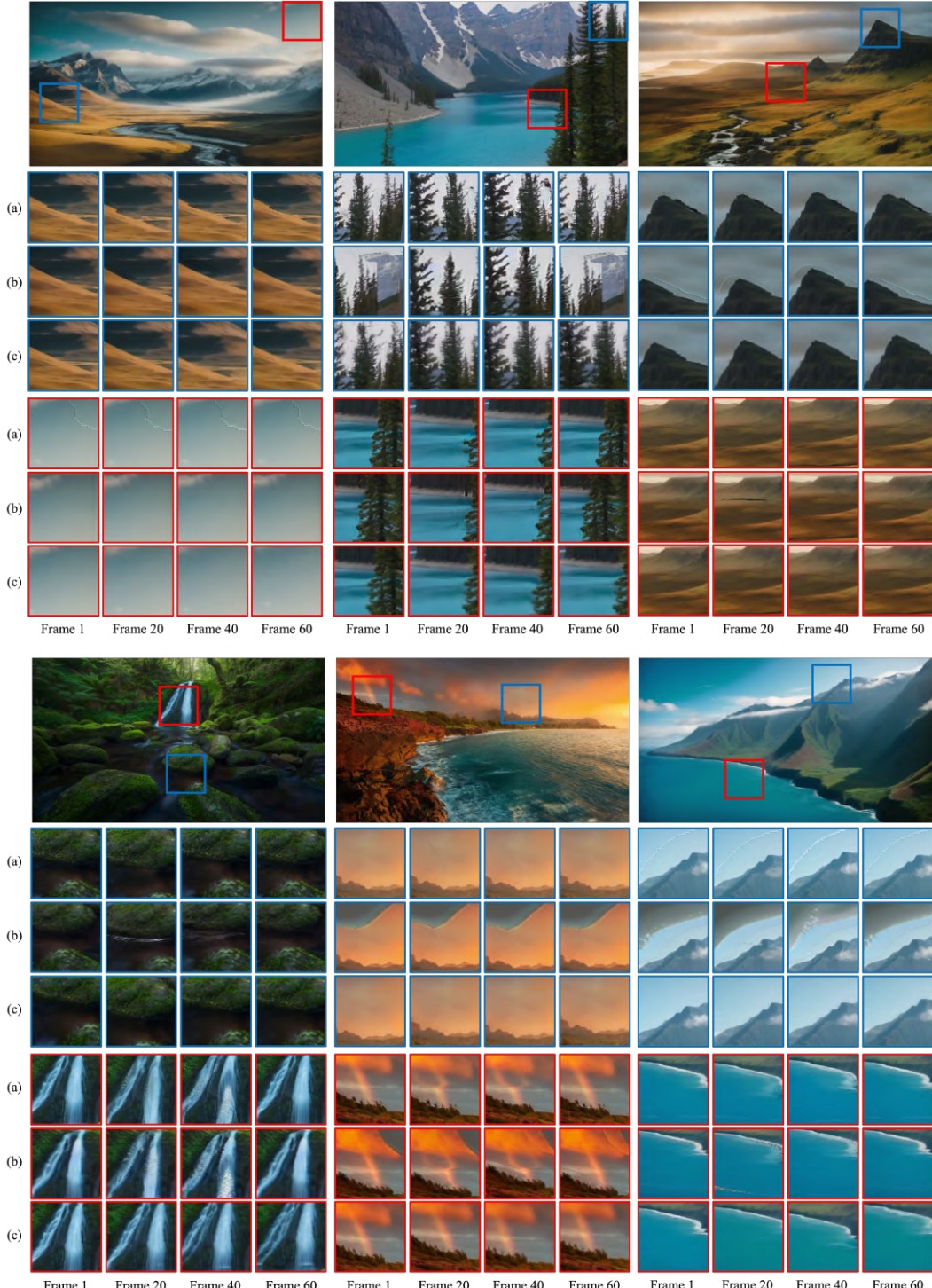

Figure 7: **"in-the-wild" dataset Results.** (a) 3D Cinemagraphy (Li et al., 2023), (b) Make-It-4D (Shen et al., 2023), (c) Ours

## B  APPLICATION: 4D SCENE GENERATION

Our framework is designed to seamlessly incorporate 3D scene generation models, facilitating straightforward spatial expansion. Fig. 8 shows the results of incorporating the 3D Scene Gener-

ation model, LucidDreamer (Chung et al., 2023), into our method. LucidDreamer converts single images into point clouds and progressively fills empty areas with an inpainting model, enabling spatiotemporal expansion when incorporated into our framework. This incorporation enables the creation of videos with more natural motion and expansive views.

Additionally, we conduct comparisons with recent work on the 4D Scene Generation model, Vivid-Dream (Lee et al., 2024). Unlike our approach, VividDream directly generates multi-view videos through the T2V model without utilizing motion estimation for temporal expansion. Fig. 9 compares the results of our framework, incorporating LucidDreamer (Chung et al., 2023), with those of VividDream (Lee et al., 2024). Our framework also integrates Viewcrafter (Yu et al., 2024), which enhances multi-view video generation, conditioned on point cloud-rendered videos in a single diffusion pass. We render the comparisons using the closest matching images and cameras available, as the code and data were not disclosed. Since VividDream (Lee et al., 2024) generates videos independently from multiple views, this approach results in motion ambiguity that leads to blurred reconstructions in fluid scenes, failing to accurately capture various motions. In contrast, our method estimates consistent 3D motion based on 2D motion, subsequently generating videos that achieve high-quality video with more natural motion.

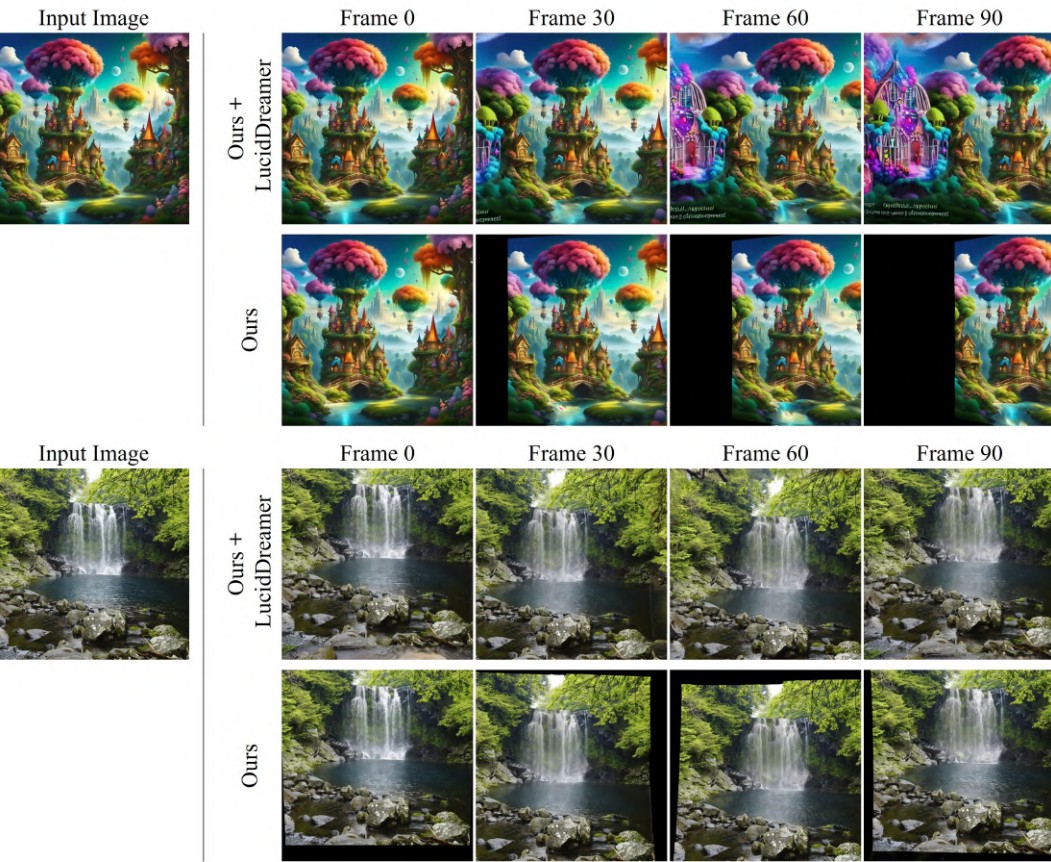

Figure 8: Results of Our framework with LucidDreamer (Chung et al., 2023)

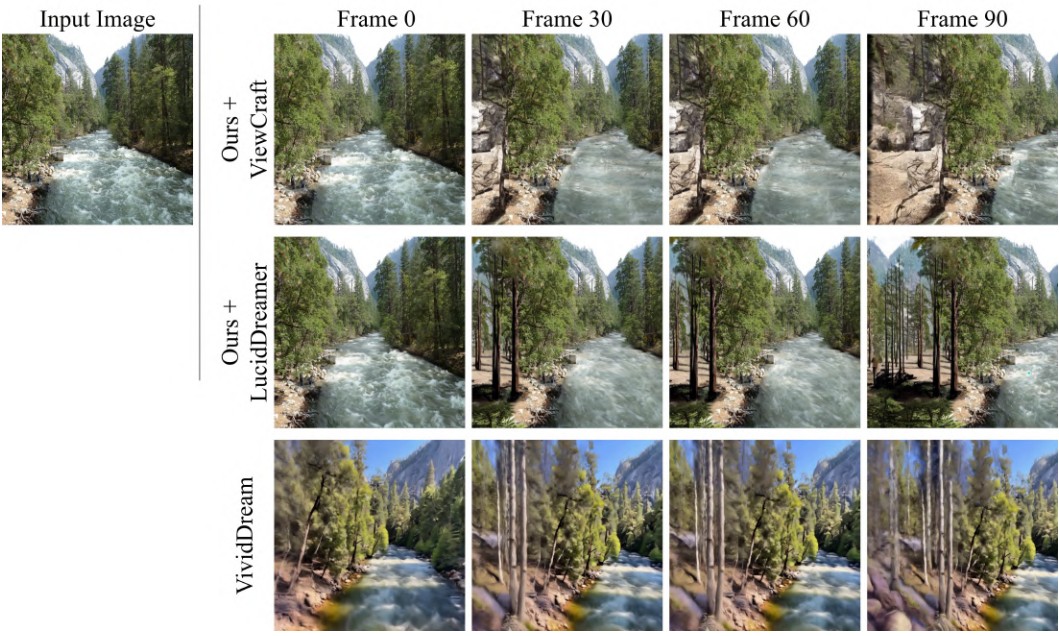

Figure 9: Comparison results of Ours and VividDream (Lee et al., 2024)

## C SINGLE IMAGE ANIMATION MODEL

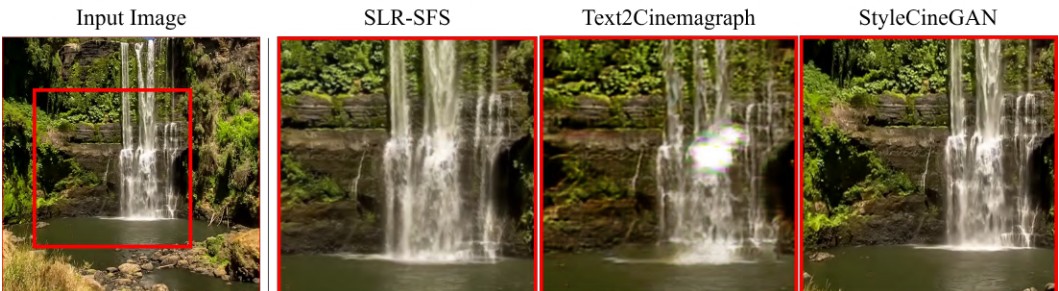

Figure 10: Comparison of rendered videos from 4D Gaussians trained by three different single image animation models. We optimized each 2D flows with 3D-MOM and trained the 4D Gaussians for comparison. By evaluating the visual results of the rendered videos, we can assess the effectiveness of the three single image animation models on our framework.

In our model, it is crucial to utilize a single image animation model that precisely estimate 2D motion from Multi-view images and generate multi-view videos by accurately reflecting 3D motion. Fig. 10 shows the results of trained 4D Gaussians using animated videos by different single image animation models, SLR-SFS (Fan et al., 2023), Text2Cinemagraph (Mahapatra et al., 2023) and StyleCineGAN (Choi et al., 2024). The Eulerian flows estimated by each model enable the generation of consistent 3D motion through 3D-MOM, which facilitates the restoration of natural motion in 4D scene. Additionally StyleCineGAN (Choi et al., 2024) can generate natural videos not only of fluids like water but also of clouds and smoke, allowing for the reconstruct various motions when utilized to our framework. The results can be seen in Fig. 11. As the field of single image animation expands, we are able to generate consistent 3D motion from various 2D motions, allowing for expansion across a wider variety of scenes.

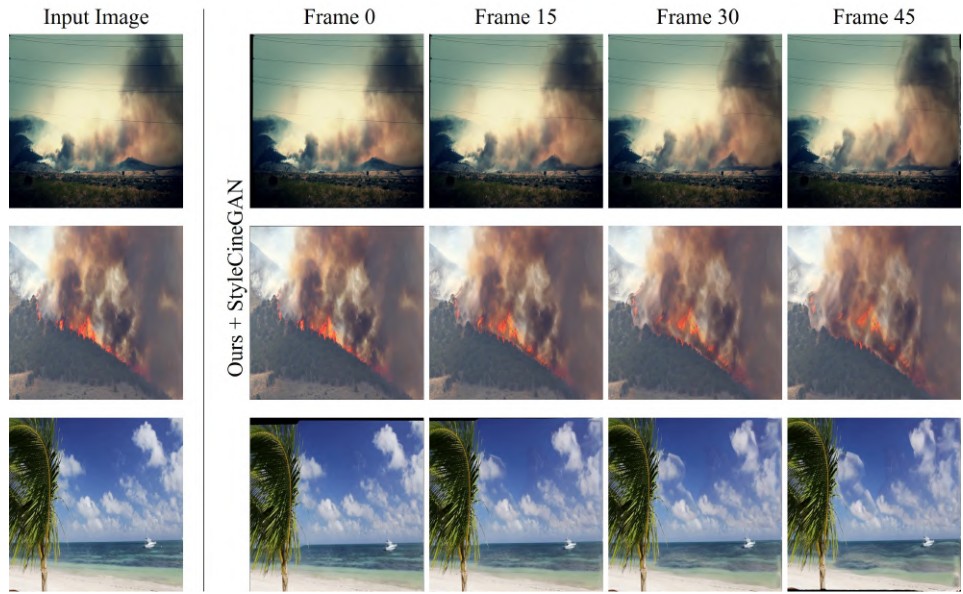

Figure 11: Results of Our framework with StyleCineGAN (Choi et al., 2024)

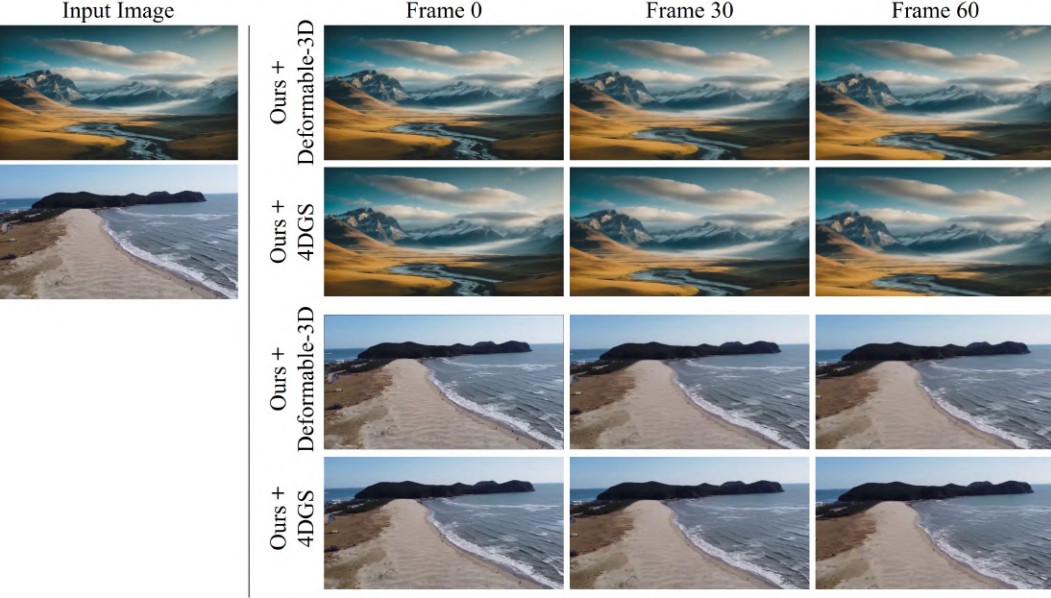

Figure 12: Results of Our framework with Deformable-3D (Yang et al., 2024)

## D  4D GAUSSIAN SPLATTING

Since our framework utilizes 4D Gaussians to model complete 3D spaces with motion, the expressiveness of the model itself significantly influences the quality of the final results. Fig. 12 shows the results of implementing the Deformable-3D (Yang et al., 2024) within our framework. Compared to previous results using 4D-GS (Wu et al., 2024), it reconstructs low-fidelity 4D scenes and generates videos with reduced realism. Therefore, by utilizing 4D-GS (Wu et al., 2024), our framework is capable of producing more immersive Dynamic Scene Videos. This experiment demonstrates the adaptability of our model, and as advancements are made in the field of 4D Gaussians, the performance of our framework also improves.

# E  EFFECT OF TWO-STAGE TRAINING

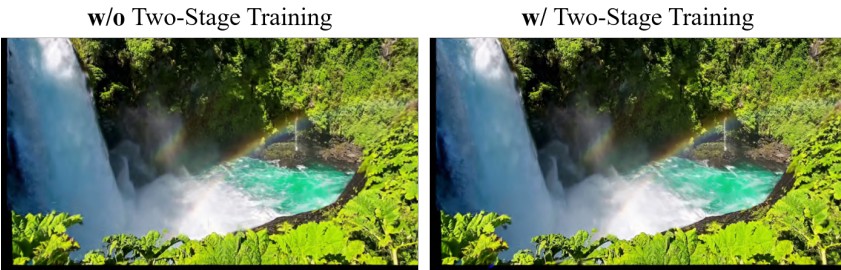

Figure 13: Effect of the two-stage training on 4D Gaussians. Comparison of our model with and without two-stage training. The left part shows results trained with videos from all viewpoints, right part shows results trained with videos from sampled viewpoints.

Table 2: Comparison of metrics and runtime with and without applying two-stage training. Through two-stage training, we significantly reduced the overall runtime while quantitatively demonstrating that the results are comparable to those trained with videos from all viewpoints.

| Method | Metrics | | | | | | Runtime | |
|---|---|---|---|---|---|---|---|---|
| | PSNR | SSIM | LPIPS | M.PSNR | M.SSIM | M.LPIPS | Step1 | Step2 |
| w/o 2-stage running | **18.17** | **0.91** | 0.18 | **22.10** | **0.96** | **0.08** | 1h 3m 39s | 51m 34s |
| w/ 2-stage running | 17.93 | 0.90 | **0.17** | 22.05 | **0.96** | **0.08** | **1m 50s** | **39m 45s** |

To achieve faster and more stable results with our algorithm, we separated the 4D Gaussians learning process by viewpoints and time axis. In step 1, we trained 3D Gaussians using all viewpoints, and in step 2, we trained 4D Gaussians using videos from sampled viewpoints.

Fig. 13 shows the results of training 4D Gaussians with animated videos for all viewpoints, while the bottom shows the results of our two-stage training approach trained on only three viewpoint videos. This demonstrates that our training method produces results almost identical to those obtained by training with videos from all viewpoints. Additionally, as shown in Table 2, which was evaluated on a sample validation set, our method not only maintains high performance but also achieves a significant efficiency improvement. It is over 30 times faster in generating videos and reduces the training time for the 4D Gaussians by a third. It demonstrates an optimal balance between speed and accuracy.

## F    LONG FRAME RESULTS

Recent diffusion-based T2V models capable of simultaneously generating multi-angle images and cinemagraphy, similar to Dynamic Scene Videos, have emerged (Shi et al., 2024), (Xing et al., 2025). However, these models experience a sharp increase in computational load with the number of frames, limiting them to a maximum of 30 frames per inference and requiring lengthy inference times for each new view. In contrast, our framework can reconstruct long durations using explicit 4D Gaussians, allowing for the creation of novel view videos in a shorter time and at a lower cost. Fig. 14 demonstrates that our framework can produce long videos maintaining natural motion and high-fidelty, capable of generating up to 330 frames. The high compatibility of our framework ensures that as the field of 4D Gaussians advances, the performance of our framework also improves, enabling the production of longer Dynamic Scene Videos.

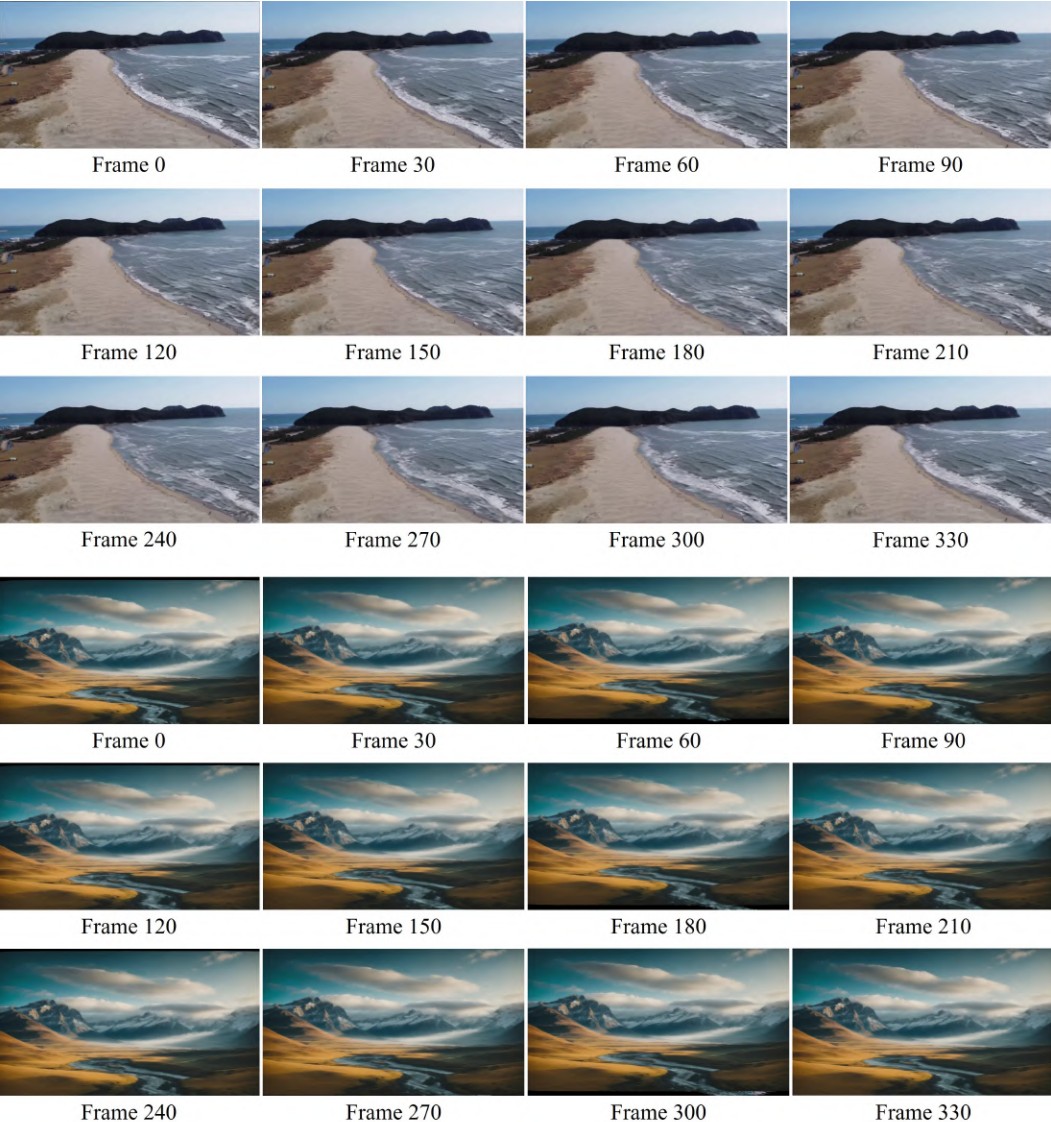

Figure 14: Qualitative results of the long frame experiment on our framework

# G  DETAILED PROCESS OF MODEL SELECTION FOR OUR FRAMEWORK

In this appendix, we provide an in-depth examination of the experimental process and the decision-making rationale involved in selecting the specific components of our framework. Our objective was to ensure a robust capability for high-fidelity 4D scene reconstruction, necessitating systematic experimentation and domain-specific insights. For depth prediction models, we evaluated ZoeDepth (Bhat et al., 2023), noted for its zero-shot accuracy and broad adaptability, and DPT (Ranftl et al., 2021), which utilizes advanced transformer technology for dense prediction tasks. ZoeDepth (Bhat et al., 2023) was selected due to its superior robustness and generalization across varied scenarios, aligning with our framework's requirements. In the realm of 2D motion estimation, we tested models including Holynski et al. (Holynski et al., 2021) approach for animating pictures with Eulerian motion fields and techniques for controllable animation of fluid elements in still images. Holynski et al. (Holynski et al., 2021) method was chosen for its ability to generate consistent, looping animations, particularly effective for natural phenomena such as fluids and clouds. Regarding video generation, our evaluations included Text2Cinemagraph (Mahapatra et al., 2023), SLR-SFS(Fan et al., 2023), and StyleCineGAN (Choi et al., 2024), with SLR-SFS (Fan et al., 2023) being selected for its excellent balance between computational efficiency and temporal coherence, essential for our framework's integration needs. Lastly, for 4D Gaussian optimization, we compared 4D-GS (Wu et al., 2024) and Deformable 3D Gaussian Splatting (Yang et al., 2024), opting for 4D-GS(Wu et al., 2024) due to its efficiency in modeling spatiotemporal dynamics while maintaining high fidelity. This methodical approach to component selection ensures that our framework is not only effective but also adaptable, paving the way for future enhancements and broad applications in dynamic scene reconstruction.

