# OpenReview forum: "Optimizing 4D Gaussians for Dynamic Scene Video from Single Landscape Images"
_ICLR.cc/2025/Conference — ICLR 2025 Poster_

### Official Review · Reviewer_WRaw · 2024-10-19

**Soundness:** 2
**Presentation:** 3
**Contribution:** 2
**Rating:** 6
**Confidence:** 4

**Summary:**

This paper presents a novel framework for generating realistic 4D dynamic videos from a single landscape image. The approach uses a three-stage optimization process: first, it optimizes 3D Gaussians by generating multi-view images from the input; second, it estimates consistent 3D motion across views using the 3D Motion Optimization Module (3D-MOM); and third, it optimizes 4D Gaussians by applying the estimated motion to model changes in position, rotation, and scaling over time. This technique outperforms existing methods, such as 3D Cinemagraphy and Make-It-4D, by producing better qualitative results with reduced artifacts. The use of Gaussian Splatting contributes to improved PSNR and sharper video effects, as demonstrated in the supplementary videos.

**Strengths:**

+ Reasonable Pipeline:

The choice of Gaussian splatting offers better depth approximation compared to Layered Depth Images (LDI), leading to fewer artifacts in novel view synthesis.

+ Natural 3D Motion Consistency:

The use of 3D consistent motions derived from 2D motion guidance results in more natural and coherent dynamics in the generated videos.

+ Improved Performance:

The method demonstrates superior PSNR and visually appealing results compared to the approaches by Shen et al. and Li et al.

**Weaknesses:**

It has several limitations, where my main concerns are the implementation details and its evaluations.

+ Gaussian Splatting Optimization:

In sec. 3.1, the explanation for both (a) camera settings and (b) multi-view optimization is unclear. Monocular depth estimation typically outputs normalized depth or disparity, which loses the depth scale. Additionally, a single image usually lacks intrinsic or extrinsic camera parameters. How were these parameters obtained?

In Eq. 3, what serves as the ground truth (GT) for multiview loss? A single image cannot provide multiple views for supervision unless additional views are available. However, the paper claims to only use a single image as input. How is Eq. 3 implemented in this case?

+ Motion Estimation:

In sec. 3.2, there is some ambiguity regarding the motion estimation process. Are you using 2D optical flow? While projecting 3D motion into a 2D image would result in 2D optical flow, which offers rich motion data, the implementation is unclear.

For various views, such as in Fig. 1, it seems semantic masks (e.g., for water or fluid) are used. First, how do you get these masks?  Second, if you render a novel view with most black regions and it to the motion estimation model, how do you get a reasonable motion estimation result? The paper lacks details on this critical point.

+ 4D Gaussian Training Process:

In sec. 3.3, the training process for 4D Gaussians is confusing. In pipeline 1 and 2, the method seems to already estimate 3D Gaussians and their motions. This suggests that you could simply deform the Gaussians to generate a video. However, training 4D Gaussians requires a video, but the input is a single 2D image.

Line 343 mentions using a video generation model to produce a pseudo-video (is this using OpenSORA?), which is then used to create a 4D Gaussian with initialized motions and 3D points, supervised by the input image and generated video. Is this correct?

+ Heavy Reliance on Pretrained Models:

If I understand correctly, the method seems to rely heavily on pretrained models. Specifically, it seems to (a) use video generation models to create a video, (b) apply depth and motion models to estimate initial depth and motion, and (c) lift these into 4D for dynamic novel view synthesis (NVS). This implies the core technique is lifting generated 4D content from 2D guidance, which could limit the novelty of the approach.

+ Applicability to Specific Motions:

The method appears limited to handling fluid motions, such as water.

+ Evaluation Metrics:

PSNR may not be a meaningful metric for evaluating motion generation. A user study could provide more insightful feedback. Nevertheless, I acknowledge that the method shows better performance than the baselines, as demonstrated

**Questions:**

See above.

---

> ### Author Response · Authors · 2024-11-25
> **Comment 1/5 for Reviewer WRaw**
>
> We thank the reviewer for the careful reading and thoughtful comments. Below, we address the reviewer’s questions and provide detailed explanations. The corresponding revisions in the manuscript are marked in blue for clarity. We hope that the responses below, along with the updated manuscript, address the reviewer’s concerns thoroughly. Due to character limits on OpenReview, our response has been divided into five parts for ease of readability. Thank you for your understanding.
>
> ---
>
> **Weakness 1:** Gaussian Splatting Optimization: In Sec. 3.1, the explanation for both (a) camera settings and (b) multi-view optimization is unclear. Monocular depth estimation typically outputs normalized depth or disparity, which loses the depth scale. Additionally, a single image usually lacks intrinsic or extrinsic camera parameters. How were these parameters obtained?
>
> **Response:** Thank you for your thoughtful question. As you noted, monocular depth estimation typically outputs normalized depth values without an absolute scale, and single images lack intrinsic or extrinsic camera parameters. Our framework addresses these challenges with practical and widely adopted solutions in the 3D Scene Generation field:
>
> - **Depth Values and 3D Lifting:** While the depth values are not scaled to real-world measurements, they provide sufficient relative information for constructing a 3D point cloud. By combining these depth values with camera parameters, we unproject the 2D image into a relative 3D space. This enables effective 3D Gaussian optimization.
>
> - **Camera Parameters:**
>   - **Intrinsic Parameters:** Determined based on the input image size, using standard conventions in computer vision.
>   - **Extrinsic Parameters:** Initialized using a predefined camera trajectory starting from an identity matrix. This approach allows us to optimize 3D Gaussians effectively, even with single-image inputs.
>
> These methods ensure that our framework overcomes the inherent limitations of monocular depth estimation and single-image camera parameterization, enabling robust 3D scene reconstruction.
> We appreciate your feedback, which has helped us improve clarity. The **Sec. 4.1.2** of the revised manuscript now provides more details on this process for transparency and understanding.
>
> ---
>
> **Weakness 2:** Gaussian Splatting Optimization: In Eq. 3, what serves as the ground truth (GT) for multiview loss? A single image cannot provide multiple views for supervision unless additional views are available. However, the paper claims to only use a single image as input. How is Eq. 3 implemented in this case?
>
> **Response:** Thank you for raising this insightful question and for highlighting an important aspect of our implementation. As outlined in Sec. 3.1, the process of generating multi-view images from a single input image builds on established methods. However, we recognize that the explanation provided in the original manuscript may not have been detailed enough, and we sincerely apologize for any confusion this may have caused.
>
> To implement Eq. 3, we generate novel views by manipulating the extrinsic camera parameters to render multi-view images from the 3D unprojection of the single input image. To address the holes introduced during rendering, we apply a simple linear interpolation, ensuring efficient and high-quality multi-view image generation. These generated multi-view images act as the ground truth (GT) for the multiview loss. This approach preserves structural consistency across the generated views, enabling the effective optimization of the 3D Gaussian representation.
>
> Your thoughtful observation prompted us to clarify this process in the revised manuscript. We have now provided a more detailed explanation to ensure transparency and better understanding of how Eq. 3 is implemented. We are deeply grateful for your constructive feedback, which has significantly contributed to improving the clarity and overall presentation of our work. Thank you again for your valuable input!

---

> ### Author Response · Authors · 2024-11-25
> **Comment 2/5 for Reviewer WRaw**
>
> **Weakness 3:** Motion Estimation: In sec. 3.2, there is some ambiguity regarding the motion estimation process. Are you using 2D optical flow? While projecting 3D motion into a 2D image would result in 2D optical flow, which offers rich motion data, the implementation is unclear.
>
> **Response:** Thank you for your insightful question. As you correctly noted, projecting 3D motion into a 2D image provides excellent 2D motion data. However, to the best of our knowledge, there is no existing method that directly estimates motion in 3D.
>
> To address this gap, we propose our core module, the 3D Motion Optimization Module (3D-MOM), which indirectly estimates consistent 3D motion by leveraging 2D motion data, such as optical flow. This approach allows us to utilize existing 2D motion estimation methods while lifting them into 3D space.
>
> Our module is not limited to the Eulerian flow model used in this framework. Thanks to its high compatibility, 3D-MOM can extend various Eulerian-based motion models into 3D, as demonstrated in **Appendix C** and [Project Page](https://gramnard.github.io/ICLR_3D_MOM/#section7). Moreover, 3D-MOM opens the door for applying the rich body of research on 2D motion estimation in single-image animation to 3D, potentially making a significant impact on the field.
>
> ---
>
> **Weakness 4:** 4D Gaussian Training Process: In sec. 3.3, the training process for 4D Gaussians is confusing. In pipeline 1 and 2, the method seems to already estimate 3D Gaussians and their motions. This suggests that you could simply deform the Gaussians to generate a video. However, training 4D Gaussians requires a video, but the input is a single 2D image. Line 343 mentions using a video generation model to produce a pseudo-video (is this using OpenSORA?), which is then used to create a 4D Gaussian with initialized motions and 3D points, supervised by the input image and generated video. Is this correct?
>
> **Response:** Thank you for your detailed question and for closely examining the 4D Gaussian training process. Your understanding is mostly correct, and we appreciate the opportunity to clarify this aspect of our work.
>
> **Pseudo-Video Generation for 4D Gaussian Training:**
> To train 4D Gaussians, multi-view video sequences are required. As you noted, the input to our framework is a single 2D image. To address this, we leverage the multi-view images synthesized earlier along with the estimated 3D motions to generate frame sequences (pseudo-videos) using single-image animation models, such as SLR-SFS. These pseudo-videos provide the necessary supervision for training the 4D Gaussian framework.
>
> **Training Process:**
> As described in Sec. 3.3, our training process consists of the following steps:
> 1. **3D Gaussian Initialization and Motion Estimation:**
>    The 3D Gaussians and their motions are first initialized using the multi-view images and the 3D Motion Optimization Module (3D-MOM). This provides a strong starting point for further refinement.
> 2. **Pseudo-Video Generation:**
>    The generated multi-view images and the initialized 3D motions are fed into a single-image animation model to produce frame sequences for sampled viewpoints.
> 3. **Two-Stage Training for 4D Gaussians:**
>    - **Spatial Training:** Multi-view images are used to optimize the 3D Gaussians, focusing on spatial consistency.
>    - **Temporal Training:** The deformation network is trained using pseudo-videos from the sampled viewpoints, enabling the representation of dynamic temporal changes in the 4D Gaussian framework. This approach avoids the need to generate videos for all views, significantly improving efficiency without compromising quality.
>
> While it is possible to deform 3D Gaussians to generate a video directly, this approach would lack the temporal coherence and detailed dynamics enabled by the explicit modeling of 4D Gaussians. The inclusion of temporal deformation in the 4D Gaussian representation ensures high-quality spatiotemporal reconstructions.
> We hope this explanation clarifies the process and highlights the unique contributions of our framework. Thank you once again for raising this important question and allowing us to refine the description in our manuscript.

---

> ### Author Response · Authors · 2024-11-25
> **Comment 3/5 for Reviewer WRaw**
>
> **Weakness 5:** Heavy Reliance on Pretrained Models: If I understand correctly, the method seems to rely heavily on pretrained models. Specifically, it seems to (a) use video generation models to create a video, (b) apply depth and motion models to estimate initial depth and motion, and (c) lift these into 4D for dynamic novel view synthesis (NVS). This implies the core technique is lifting generated 4D content from 2D guidance, which could limit the novelty of the approach.
>
> **Response:** Thank you for your valuable feedback and the opportunity to clarify our contributions. While it is true that our framework employs pretrained models for foundational tasks such as depth and motion estimation, the core novelty lies in how we integrate and extend these components to tackle the underexplored challenge of consistent dynamic scene reconstruction.
>
> - **Core Novelty:**
>
> 1. **3D Motion Optimization Module (3D-MOM):**
>    The 3D-MOM bridges the gap between 2D motion estimation and dynamic scene reconstruction by lifting 2D motion into consistent 3D motion. Unlike existing methods, which rely on motion models directly, our approach ensures spatiotemporal coherence across views. This modular design allows our framework to adapt to various single-image animation techniques, enhancing its applicability across diverse dynamic content.
>
> 2. **Extending to 4D Scene Generation:**
>    Our framework introduces 4D scene generation as a novel task, achieved by combining prior 3D spatial scene generation techniques with our dynamic scene reconstruction approach. By expanding orthogonally into the temporal domain, we optimize temporal dynamics within Gaussian representations, enabling cohesive spatiotemporal generations that surpass the limitations of purely spatial models. This new application, detailed in Sec. 4.5, highlights the transformative potential of integrating 3D and dynamic scene techniques into a unified framework.
>
> While pretrained models support foundational tasks (e.g., depth or motion estimation), they are tools within our larger framework. The novelty lies in:
>    - **Strategic Integration:** Our approach leverages these models to achieve robust spatiotemporal consistency in 4D outputs.
>    - **Modularity and Scalability:** The framework’s design ensures adaptability to future advances in pretrained models, enabling continued improvements in performance and applicability.
>
> This work demonstrates that by integrating pretrained models with innovative modules like 3D-MOM, we can address core challenges in dynamic scene reconstruction. Our approach sets the stage for scalable and adaptable 4D scene generation, contributing a meaningful advancement to the field.
> We greatly appreciate your feedback, which has helped us better articulate the contributions and impact of our work.

---

> ### Author Response · Authors · 2024-11-25
> **Comment 4/5 for Reviewer WRaw**
>
> **Weakness 6:** Applicability to Specific Motions: The method appears limited to handling fluid motions, such as water.
>
> **Response:** Eulerian flow is a concept from fluid dynamics commonly used in single-image animation methods. For example, Holynski et al. [1] and subsequent works have leveraged Eulerian flow to generate diverse looping videos. Recently, these methods have been extended beyond fluid motion to other domains, such as clouds [2] and clothing motion [3]. **In Appendix C** and [Project Page](https://gramnard.github.io/ICLR_3D_MOM/#section7), we demonstrate that by replacing Holynski et al. [1] with another Eulerian flow-based model [2] in our framework, the 3D-MOM module still performs effectively, generating consistent 3D motion.
>
> Furthermore, our framework is designed to be broadly compatible with any single-image animation model capable of animating specified regions through masks. Handling highly complex motions, such as human or articulated movements, presents inherent challenges due to the current limitations of single-image animation techniques. While recent diffusion-based methods show promise in generating diverse and intricate motions, they struggle to maintain spatiotemporal consistency, particularly across multi-view settings. As a result, our work focuses on a fundamental aspect of dynamic scene reconstruction: representing natural phenomena, such as fluids, clouds, and dynamic textures, which are essential for applications in visual effects, virtual environments, and immersive media.
>
> As single-image animation techniques continue to evolve and expand to broader content domains, our proposed approach stands ready to enable consistent 3D motion estimation across diverse content types. The modular nature of our framework ensures its adaptability, facilitating the reconstruction of 4D scenes and broadening its applicability to increasingly complex motion domains in the future.
>
> **References:**
>
> [1] Aleksander Holynski, Brian L Curless, Steven M Seitz, and Richard Szeliski. *Animating pictures with Eulerian motion fields.* In Proceedings of the IEEE/CVF Conference on Computer Vision and Pattern Recognition, pp. 5810–5819, 2021.
> [2] Choi, Jongwoo, et al. *StyleCineGAN: Landscape Cinemagraph Generation using a Pre-trained StyleGAN.* Proceedings of the IEEE/CVF Conference on Computer Vision and Pattern Recognition, 2024.
> [3] Bertiche, Hugo, et al. *Blowing in the wind: Cyclenet for human cinemagraphs from still images.* Proceedings of the IEEE/CVF Conference on Computer Vision and Pattern Recognition, 2023.

---

> ### Author Response · Authors · 2024-11-25
> **Comment 5/5 for Reviewer WRaw**
>
> **Weakness 7:** Evaluation Metrics:  PSNR may not be a meaningful metric for evaluating motion generation. A user study could provide more insightful feedback. Nevertheless, I acknowledge that the method shows better performance than the baselines, as demonstrated.
>
> **Response:**
>
> Thank you for your thoughtful comment and for acknowledging the performance of our method compared to the baselines. To address your concern about evaluation metrics, we conducted an additional user study to provide more comprehensive and insightful feedback on motion generation quality. This study further demonstrated that our model outperforms state-of-the-art (SOTA) methods in the Dynamic Scene Video domain across multiple aspects.
> The results of this user study have been incorporated into **Table 1: Quantitative Results**, providing further evidence of the robustness and quality of our approach.
>
> We appreciate your suggestion, which allowed us to better highlight the strengths of our approach.
>
> | **Method**             | **PSNR ↑**     | **SSIM ↑**     | **LPIPS ↓**    | **PIQE ↓**       | **Immersion (%)** | **Realism (%)** | **Structural Consistency (%)** | **Quality (%)** |
> |-------------------------|----------------|----------------|----------------|------------------|--------------------|------------------|-------------------------------|------------------|
> | DynamiCrafter [1]       | 14.98          | 0.81           | 0.23           | 24.58            | -                  | -                | -                             | -                |
> | Motion-I2V [2]          | 14.38          | 0.80           | 0.31           | 8.40             | -                  | -                | -                             | -                |
> | 3D-Cinemagraphy [3]     | 17.30          | 0.83           | 0.17           | 8.93             | 31.87             | 31.87           | 28.75                        | 30.31           |
> | Make-It-4D [4]          | 16.98          | 0.81           | 0.20           | 8.30             | 11.87             | 10.31           | 8.43                         | 9.06            |
> | **Ours**                | **20.57**      | **0.90**       | **0.14**       | **7.80**         | **56.25**         | **57.81**       | **62.81**                     | **60.25**       |
>
>
>
>
> **References:**
>
> [1] Xing, Jinbo, et al. *Dynamicrafter: Animating open-domain images with video diffusion priors.* European Conference on Computer Vision. Springer, Cham, 2025.
> [2] Shi, Xiaoyu, et al. *Motion-I2V: Consistent and controllable image-to-video generation with explicit motion modeling.* ACM SIGGRAPH 2024 Conference Papers. 2024.
> [3] Li, Xingyi, et al. *3D Cinemagraphy from a Single Image.* Proceedings of the IEEE/CVF Conference on Computer Vision and Pattern Recognition. 2023.
> [4] Shen, Liao, et al. *Make-it-4D: Synthesizing a Consistent Long-Term Dynamic Scene Video from a Single Image.* Proceedings of the 31st ACM International Conference on Multimedia. 2023.

---

> > ### Author Response · Authors · 2024-12-01
> >
> > **Dear Reviewer WRaw,**
> >
> > We deeply appreciate your thoughtful and constructive feedback throughout this review process. As the discussion period is nearing its conclusion, we wanted to follow up to ensure that we have fully addressed your concerns and to provide additional updates that we believe further strengthen the contributions and clarity of our work.
> >
> > In response to your valuable suggestions, we conducted a user study to evaluate the realism, immersion, and structural consistency of our motion generation. The study results, now included in Table 1 of the manuscript, highlight that our method outperforms state-of-the-art approaches across key metrics. This provides further evidence of the robustness and quality of our framework.
> >
> > Additionally, we conducted experiments on dynamic phenomena such as clouds and smoke to demonstrate the adaptability of our framework to broader content domains. These results validate the versatility of our approach in handling fluid-like dynamics beyond traditional liquids, addressing the concern of specificity. We have updated our [project page](https://gramnard.github.io/ICLR_3D_MOM/) with videos showcasing these new results and invite you to review them.
> >
> > While our framework leverages pretrained models for foundational tasks, the core novelty lies in the integration of these components into a unified pipeline. The 3D Motion Optimization Module (3D-MOM) bridges the gap between 2D motion estimation and 3D scene reconstruction, enabling consistent and dynamic motion generation. Additionally, our work introduces 4D Gaussian scene generation as a novel task, emphasizing its potential impact on spatiotemporal modeling.
> >
> > We are sincerely grateful for your insights, which have been instrumental in refining and strengthening our work. If you have any further suggestions or concerns, we would be honored to address them promptly. Thank you once again for your time, careful review, and consideration of our submission.
> >
> > **Best regards.**

---

> > > ### Comment · Reviewer_WRaw · 2024-12-02
> > >
> > > I thank the authors for their detailed feedback, which must have required significant effort. Overall, the rebuttal has addressed most of my concerns. However, I still have two points that I hope the authors can further clarify:
> > >
> > > a) Chicken-and-egg problem: I understand that after unprojection, the Gaussian points become 3D representations that can be rendered into novel views. However, this approach seems to resemble a chicken-and-egg problem. If novel views are created this way, wouldn’t the best 3D reconstruction achievable simply replicate the input mono-depth? What's the 3D scene you want to reconstruct? Are you suggesting that the depth is fixed and that only other Gaussian attributes, such as motions, are optimized?
> > >
> > > > Weakness 2: Gaussian Splatting Optimization: In Eq. 3, what serves as the ground truth (GT) for multiview loss? A single image cannot provide multiple views for supervision unless additional views are available. However, the paper claims to only use a single image as input. How is Eq. 3 implemented in this case?
> > >
> > > > Response: Thank you for raising this insightful question and for highlighting an important aspect of our implementation. As outlined in Sec. 3.1, the process of generating multi-view images from a single input image builds on established methods. However, we recognize that the explanation provided in the original manuscript may not have been detailed enough, and we sincerely apologize for any confusion this may have caused.
> > >
> > > > To implement Eq. 3, we generate novel views by manipulating the extrinsic camera parameters to render multi-view images from the 3D unprojection of the single input image. To address the holes introduced during rendering, we apply a simple linear interpolation, ensuring efficient and high-quality multi-view image generation. These generated multi-view images act as the ground truth (GT) for the multiview loss. This approach preserves structural consistency across the generated views, enabling the effective optimization of the 3D Gaussian representation.
> > >
> > > b) Code availability: Will the code be released? I imagine that such a complex pipeline would be challenging to reproduce. Code release would undoubtedly help the community and facilitate future research.
> > >
> > > Despite my remaining concerns, I am satisfied with the results presented on the webpage and greatly appreciate the authors’ hard work during the rebuttal period. Therefor, I will raise my score to 6.

---

> > > > ### Author Response · Authors · 2024-12-02
> > > >
> > > > **Dear Reviewer WRaw,**
> > > >
> > > > Thank you again for your thoughtful feedback and for raising such insightful questions. Your observations have been invaluable in helping us clarify key aspects of our work, and we deeply appreciate the time and effort you have devoted to reviewing our submission.
> > > >
> > > > ----
> > > >
> > > > **Question 1:** Chicken-and-egg problem: I understand that after unprojection, the Gaussian points become 3D representations that can be rendered into novel views. However, this approach seems to resemble a chicken-and-egg problem. If novel views are created this way, wouldn’t the best 3D reconstruction achievable simply replicate the input mono-depth? What's the 3D scene you want to reconstruct? Are you suggesting that the depth is fixed and that only other Gaussian attributes, such as motions, are optimized?
> > > >
> > > > **Response:** Thank you for your thoughtful feedback and for raising such insightful questions. Your observations have allowed us to further clarify the intent and scope of our method, particularly in addressing the “chicken-and-egg” concern.
> > > >
> > > > You have raised an excellent point regarding the potential limitations of relying on monocular depth estimation and whether our approach could be constrained to replicating the input mono-depth. While it is true that the initial depth estimation from a single image provides only relative depth information, our framework works within these limitations to optimize additional Gaussian attributes, such as motion, opacity, and spatial distribution.
> > > >
> > > > The goal of **Dynamic Scene Video** focuses on creating immersive, dynamic scenes that blend parallax effects, temporal motion, and consistent multi-view rendering, rather than prioritizing geometrically accurate 3D reconstructions. The monocular depth estimation provides a foundation for relative spatial consistency, which is then refined through our Gaussian optimization process to generate visually coherent and immersive dynamic outputs. By incorporating motion attributes, our method transforms static images into compelling 4D representations, offering a novel contribution to the field of dynamic scene generation.
> > > >
> > > > As highlighted in the manuscript, other studies in the **Dynamic Scene Video** domain, such as **3D-Cinemagraphy (CVPR 2023)** and **Make-It-4D (ACM MM 2023)**, also utilize monocular depth estimation as a foundation. However, these methods rely on **Layered Depth Images (LDIs)** to construct novel views, while our approach does not use LDIs. Instead, we optimize Gaussian attributes directly to achieve consistent spatial representations. While the estimated depth in these approaches may not perfectly represent the true geometry of the scene, it ensures sufficient spatial consistency to create convincing parallax effects and an immersive viewing experience. Our method builds upon these foundations by integrating temporal motion and optimizing Gaussian attributes, resulting in more dynamic and immersive visualizations.
> > > >
> > > > We also recognize that this setup presents opportunities for further exploration. Incorporating more robust depth estimation techniques could significantly enhance the realism and scalability of our approach in future iterations. This exciting avenue aligns with our commitment to advancing the field of dynamic scene reconstruction.
> > > >
> > > > ----
> > > > **Question 2:** Code availability: Will the code be released? I imagine that such a complex pipeline would be challenging to reproduce. Code release would undoubtedly help the community and facilitate future research.
> > > >
> > > > **Response:** Regarding your question about code availability, we are committed to ensuring the reproducibility of our work and supporting the research community. Upon acceptance of the paper, we will promptly release the complete implementation of our framework, including detailed training scripts and configurations, comprehensive test pipelines with benchmarking support, pretrained weights for quick validation and experimentation, and thoroughly documented instructions with clear examples to facilitate ease of use. We hope this will empower other researchers to build upon our work and foster further advancements in this area.
> > > >
> > > > We sincerely hope this response addresses your remaining concerns and highlights the potential and adaptability of our approach. Your detailed feedback has been invaluable, and we deeply appreciate your engagement throughout the review process. Thank you again for your time and consideration.
> > > >
> > > > **Best regards.**

---

### Official Review · Reviewer_NyCT · 2024-10-26

**Soundness:** 4
**Presentation:** 3
**Contribution:** 2
**Rating:** 8
**Confidence:** 4

**Summary:**

This paper introduces a novel approach to optimize 4D Gaussian Splatting representations using single landscape images. By leveraging Eulerian motion fields, it effectively generates animations of dynamic scenes like waterfalls, beaches, and rivers. The authors also present the 3D-MOM method, which addresses frame inconsistency to ensure smooth animations. Experimental results demonstrate the method’s state-of-the-art quality.

**Strengths:**

1. The authors propose a novel method to animate 3D scenes with the flow-based method. It is quite novel and can fit the landscape generation properly.

2. The results seem pretty good in certain demos and demonstrate the SOTA quality.

3. 3D-MOM method is a good balance  when merging different 2D motion in multi-view images.

**Weaknesses:**

1. **Section 3.1 Citation:** It seems that a citation for 4D Gaussian Splatting (Wu et al., CVPR 2024) is missing in Section 3.1.

2. **Comparison with VividDream:** I recommend that the authors include discussions or comparisons with VividDream (Lee et al., ArXiv 2024). Using a single image for initialization may not be robust enough, so it would be beneficial to expand the method to accommodate larger 3D scenes. Consider the motivations of viewcrafter/VividDream4D, as merely supporting "zoom in" functionality is insufficient. It’s essential to address additional capabilities like "turn left/right" or "zoom out/up/down."

3. **Multiview Generation and Novel Content:** Since only one image is used as input, the authors suggest that warping techniques can generate multiview images, but this approach may lack **novel content** and leave many empty regions. How do the authors handle these issues? If not resolved, I recommend using viewcrafter to enhance quality. Additionally, how are rendering paths chosen? Please provide more details.

**Questions:**

**Animating Pictures with Eulerian Motion Fields:** This technique primarily supports fluid motion, such as water and smoke. How does it handle other types of motion, like wind or more complex movements? I am curious about the potential future improvements of this method. Could it be adapted for human or animated motions in the future?

 **Handling $M_0$:** How is $M_0$ obtained? Is SAM or another method used?

Overall, I think the demo and the presentation of the paper are good. My main consideration lies in weakness 2,3,4.

Besides, I just think working on landscape images is quite small (not quite confident).  if the authors can demonstrate this pipeline could be applied in 4D scene generation, I would like to highly recommend accepting this paper.

---

> ### Author Response · Authors · 2024-11-25
> **Comment 1/3 for Reviewer NyCT**
>
> We thank the reviewer for the careful reading and thoughtful comments. Below, we address the reviewer’s questions and provide detailed explanations. The corresponding revisions in the manuscript are marked in blue for clarity. We hope that the responses below, along with the updated manuscript, address the reviewer’s concerns thoroughly. Due to character limits on OpenReview, our response has been divided into three parts for ease of readability. Thank you for your understanding.
>
> ---
>
> **Weakness 1:** Section 3.1 Citation: It seems that a citation for 4D Gaussian Splatting (Wu et al., CVPR 2024) is missing in Section 3.1.
>
> **Response:** Thank you for pointing this out. Based on your suggestion, we have added the citation for 4D Gaussian Splatting (Wu et al., CVPR 2024) in Section 3.1.
>
> ---
>
> **Weakness 2:** Comparison with VividDream: I recommend that the authors include discussions or comparisons with VividDream (Lee et al., ArXiv 2024). Using a single image for initialization may not be robust enough, so it would be beneficial to expand the method to accommodate larger 3D scenes. Consider the motivations of viewcrafter/VividDream4D, as merely supporting "zoom in" functionality is insufficient. It’s essential to address additional capabilities like "turn left/right" or "zoom out/up/down."
>
> **Response:** Thank you for pointing out this related work, which we had not previously considered and acknowledge was developed concurrently with our work. After reviewing the VividDream [1] paper, we found that it addresses a similar task by reconstructing a 3D scene from a single image and generating multi-view videos to extend into 4D. Their approach focuses on stable video diffusion-based algorithms for video generation.
>
> However, VividDream does not explicitly estimate 3D motion; instead, it directly generates multi-view videos for learning 4D Gaussians, which may result in less natural motion due to the lack of explicit 3D motion modeling. In contrast, our method leverages the well-established single-image animation domain to estimate 2D motion and then lifts it into consistent 3D motion, which is the core contribution of our framework.
>
> Our framework’s compatibility with existing 3D Scene Generation models allows it to be extended with generative models like LucidDreamer [2] or ViewCrafter [3] to support immersive 4D scene generation. This flexibility enables additional capabilities such as “turn left/right” or “zoom out/up/down,” which you mentioned, further enhancing the scope of our approach.
>
> To compare with VividDream, we incorporated LucidDreamer into our algorithm and conducted a qualitative comparison using similar scenes and camera movements, as their code is not publicly available. As detailed in **Appendix B** and [Project Page](https://gramnard.github.io/ICLR_3D_MOM/#section5), our method produces higher-quality videos with significantly more natural motion. These results have also been added to the project page for reference. Additionally, we have added a discussion of VividDream to the Related Works section.
>
> **References:**
>
> [1] Lee, Yao-Chih, et al. *VividDream: Generating 3D Scene with Ambient Dynamics.* arXiv preprint arXiv:2405.20334 (2024).
> [2] Chung, Jaeyoung, et al. *Luciddreamer: Domain-free generation of 3d gaussian splatting scenes.* arXiv preprint arXiv:2311.13384 (2023).
> [3] Yu, Wangbo, et al. *Viewcrafter: Taming video diffusion models for high-fidelity novel view synthesis.* arXiv preprint arXiv:2409.02048 (2024).

---

> ### Author Response · Authors · 2024-11-25
> **Comment 2/3 for Reviewer NyCT**
>
> **Weakness 3:** Multiview Generation and Novel Content: Since only one image is used as input, the authors suggest that warping techniques can generate multiview images, but this approach may lack novel content and leave many empty regions. How do the authors handle these issues? If not resolved, I recommend using ViewCrafter to enhance quality. Additionally, how are rendering paths chosen? Please provide more details.
>
> **Response:** Thank you for your excellent suggestion. Our task focuses on generating 4D content from a single image, which inherently cannot create entirely novel content. As a result, when rendering a wide view, it is indeed possible for empty regions to appear.
>
> However, the proposed method is designed with extensibility as a priority, allowing seamless integration with existing 3D Scene Generation models. To address the issue of novel content, we experimented with integrating our algorithm with the ViewCrafter [1] and LucidDreamer [2] models you suggested. These integrations enable the generation of novel content to fill empty regions while maintaining consistency with the original scene. The results of these experiments, including comparative analyses, have been added to the **Appendix B** and the [Project Page](https://gramnard.github.io/ICLR_3D_MOM/#section5) for reference.
>
> Regarding rendering paths, our method currently utilizes simple paths commonly used in dynamic scene videos, such as zoom-in, circular, lateral, and up-down movements. However, incorporating advanced techniques from models like ViewCrafter allows us to further expand these paths, enabling more dynamic and realistic rendering. We appreciate your recommendation, as it provides valuable direction for further refinement and improvement.
>
> **References:**
>
> [1] Yu, Wangbo, et al. *Viewcrafter: Taming video diffusion models for high-fidelity novel view synthesis.* arXiv preprint arXiv:2409.02048 (2024).
> [2] Chung, Jaeyoung, et al. *Luciddreamer: Domain-free generation of 3d gaussian splatting scenes.* arXiv preprint arXiv:2311.13384 (2023).
>
> ---
>
> **Question 1:** Animating Pictures with Eulerian Motion Fields: This technique primarily supports fluid motion, such as water and smoke. How does it handle other types of motion, like wind or more complex movements? I am curious about the potential future improvements of this method. Could it be adapted for human or animated motions in the future?
>
> **Response:** Eulerian flow is a concept from fluid dynamics commonly used in single-image animation methods. For example, Holynski et al. [1] and subsequent works have leveraged Eulerian flow to generate diverse looping videos. Recently, these methods have been extended beyond fluid motion to other domains, such as clouds [2] and clothing motion [3]. **In Appendix C** and [**Project Page**](https://gramnard.github.io/ICLR_3D_MOM/#section7), we demonstrate that by replacing Holynski et al. [1] with another Eulerian flow-based model [2] in our framework, the 3D-MOM module still performs effectively, generating consistent 3D motion.
>
> Furthermore, our framework is designed to be broadly compatible with any single-image animation model capable of animating specified regions through masks. Handling highly complex motions, such as human or articulated movements, presents inherent challenges due to the current limitations of single-image animation techniques. While recent diffusion-based methods show promise in generating diverse and intricate motions, they struggle to maintain spatiotemporal consistency, particularly across multi-view settings. As a result, our work focuses on a fundamental aspect of dynamic scene reconstruction: representing natural phenomena, such as fluids, clouds, and dynamic textures, which are essential for applications in visual effects, virtual environments, and immersive media.
>
> As single-image animation techniques continue to evolve and expand to broader content domains, our proposed approach stands ready to enable consistent 3D motion estimation across diverse content types. The modular nature of our framework ensures its adaptability, facilitating the reconstruction of 4D scenes and broadening its applicability to increasingly complex motion domains in the future.
>
> **References:**
>
> [1] Aleksander Holynski, Brian L Curless, Steven M Seitz, and Richard Szeliski. *Animating pictures with Eulerian motion fields.* In Proceedings of the IEEE/CVF Conference on Computer Vision and Pattern Recognition, pp. 5810–5819, 2021.
> [2] Choi, Jongwoo, et al. *StyleCineGAN: Landscape Cinemagraph Generation using a Pre-trained StyleGAN.* Proceedings of the IEEE/CVF Conference on Computer Vision and Pattern Recognition, 2024.
> [3] Bertiche, Hugo, et al. *Blowing in the wind: Cyclenet for human cinemagraphs from still images.* Proceedings of the IEEE/CVF Conference on Computer Vision and Pattern Recognition, 2023.

---

> ### Author Response · Authors · 2024-11-25
> **Comment 3/3 for Reviewer NyCT**
>
> **Question 2:** Handling M0: How is M0 obtained? Is SAM or another method used?
>
> **Response:** The mask was created manually using a user interface tool like [LabelMe](https://github.com/wkentaro/labelme), which is commonly used in the single-image animation domain. However, it is also possible to obtain the mask using segmentation models like SAM, as you suggested. We have added this information to the **Implementation Details** section of the revised paper.
>
> ---
>
> **Question 3:** Overall, I think the demo and the presentation of the paper are good. My main consideration lies in weakness 1, 2, 3. Besides, I just think working on landscape images is quite small (not quite confident). If the authors can demonstrate this pipeline could be applied in 4D scene generation, I would like to highly recommend accepting this paper.
>
> **Response:** Thank you for your insightful suggestion. Based on your feedback, we explored extending our framework to demonstrate its applicability in 4D scene generation. Specifically, we integrated our method with ViewCrafter [1] and LucidDreamer [2], showcasing that our framework is not only capable of achieving high performance in 4D scene reconstruction but also compatible with existing 3D Scene Generation algorithms. We have added a new section, **Sec. 4.5 Application: 4D Scene Generation**, to provide detailed insights into this extension, and all results have been uploaded to the [project page](https://gramnard.github.io/ICLR_3D_MOM/#section5).
>
> The primary strength of our proposed algorithm lies in its compatibility and flexibility. By utilizing a module that lifts 2D motion into consistent 3D motion, our framework enables seamless integration with a variety of models without requiring significant modifications. Through our experiments, we validated that combining our framework with state-of-the-art 3D Scene Generation models can extend its capabilities to immersive 4D scene generation. Additionally, our video generation experiments confirmed that the framework supports diverse content generation based on the single-image animation model used.
>
> We believe that as single-image animation techniques and generative models continue to advance, our framework will evolve alongside these developments, further improving its performance and broadening its applicability. We are deeply grateful for your invaluable suggestion and encouragement, which motivated us to explore and demonstrate the broader impact and scalability of our approach. Your insightful comment about showcasing the pipeline’s potential for 4D scene generation inspired us to push the boundaries of our work, and we sincerely thank you for your constructive feedback and support.
>
> **References:**
>
> [1] Yu, Wangbo, et al. "Viewcrafter: Taming video diffusion models for high-fidelity novel view synthesis." arXiv preprint arXiv:2409.02048 (2024).
> [2] Chung, Jaeyoung, et al. "Luciddreamer: Domain-free generation of 3d gaussian splatting scenes." arXiv preprint arXiv:2311.13384 (2023).

---

> > ### Comment · Reviewer_NyCT · 2024-11-25
> >
> > Dear authors,
> >
> > Thanks you for your comprehensive comments, all of my questions are addressed. And authors successfully demonstrates that this method could be applied in 4D scene generation, I would like to keep my positive score.

---

> > > ### Author Response · Authors · 2024-11-25
> > >
> > > **Dear Reviewer NyCT,**
> > >
> > > Thank you for your encouraging feedback and for acknowledging that our method successfully demonstrates its applicability to 4D scene generation. We are deeply grateful for your thoughtful suggestions and engagement throughout this process.
> > >
> > > We noticed that, despite your earlier comment expressing a willingness to “highly recommend accepting this paper” if the 4D scene generation application was demonstrated, the score remains unchanged. Could you kindly let us know if there are any remaining concerns or aspects we could further clarify or improve to better meet your expectations?
> > >
> > > We truly appreciate your time and support, and we are eager to address any remaining questions you might have.
> > >
> > > **Best regards.**

---

> > > > ### Comment · Reviewer_NyCT · 2024-11-28
> > > > **Change my score to accept**
> > > >
> > > > Dear Authors,
> > > >
> > > > Thank you for the reminder. After reviewing the paper and considering the feedback from other authors, I would like to change my score to "accept."
> > > >
> > > > I appreciate the contributions made and the results presented in the paper. I believe the novelty is not a limitation, and the overall contribution and outcomes of the study are strong enough to warrant acceptance.
> > > >
> > > > Best regards,

---

> > > > > ### Author Response · Authors · 2024-11-28
> > > > >
> > > > > **Dear Reviewer NyCT,**
> > > > >
> > > > > Thank you for your thoughtful feedback and for recognizing the contributions and strengths of our paper. We deeply appreciate your time, effort, and encouragement throughout this review process. Your support and acknowledgment of the applicability of our method to 4D scene generation mean a lot to us.
> > > > >
> > > > > We are also grateful for the opportunity to address the insightful comments provided by all reviewers. Through the revisions, we’ve further demonstrated the robustness, flexibility, and practical applications of our approach. Additional results and experiments, including video demonstrations and comparative analyses, are now available on our [project page](https://gramnard.github.io/ICLR_3D_MOM/#sectionRebuttal).
> > > > >
> > > > > Once again, thank you for your positive evaluation, which motivates us to continue advancing research in this area. If there are any further suggestions or questions, we’d be delighted to address them.
> > > > >
> > > > > **Best regards,**

---

### Official Review · Reviewer_aadJ · 2024-10-28

**Soundness:** 3
**Presentation:** 2
**Contribution:** 2
**Rating:** 6
**Confidence:** 4

**Summary:**

This paper proposes a method based on 4DGS to represent landscape 4D scenes from a single landscape image. It adapts Gaussian Splatting techniques for content with fluidity in the landscape images from the following aspects:
-  3DGS methods require sparse points as initiations, while this paper uses existing depth prediction models to lift 2D pixels.
- This paper proposes a module named 3D Motion Optimization to generate 3D animations with off-the-shelf 2D motion methods.
It also conducts experiments over multiple datasets to show quantitative and qualitative comparisons over existing methods.

**Strengths:**

1. Application to a new domain

This paper builds a solution to the problem of modeling landscape 4D scenes.

2. Adpation of 3DGS / 4DGS

This paper adapts 3DGS from the initializations and 4D GS with motion prior.

3. Comparison to existing methods

The method shows better results compared to the existing methods.

**Weaknesses:**

1) The problem of generating Gaussians from landscape images with fluid content is useful but seems a bit too specific/narrow. If this paper can handle general 4D content, it would have a very large impact.
2) Lack of theoretical innovation for ICLR. To address the task of fluid motion, this paper uses constant speed and direction. As 3DGS is a particle-like model, the theoretical contribution should be greater if this paper can present or show different physical-based models for fluids.
3) Presentaiton of figures. Figure 2’s motion content is hard to see. Too many small arrows and small squares hurt the quality of the figure.

**Questions:**

1) What is the max temporal length the proposed method can achieve with good quality?
It has potential value if this paper can handle a long temporal range (with generated content).

2) Can authors provide more technical insights?
This paper builds upon a lot of exciting techs (depth prediction model, 2D motion models). I would like to know if the authors selected these methods by experiments or random or with insights. If this paper can improve this part, it can benefit future work in this field.

---

> ### Author Response · Authors · 2024-11-25
> **Comment 1/3 for Reviewer aadJ**
>
> We thank the reviewer for the careful reading and thoughtful comments. Below, we address the reviewer’s questions and provide detailed explanations. The corresponding revisions in the manuscript are marked in blue for clarity. We hope that the responses below, along with the updated manuscript, address the reviewer’s concerns thoroughly. Due to character limits on OpenReview, our response has been divided into three parts for ease of readability. Thank you for your understanding.
>
> ---
>
> **Weakness 1:** The problem of generating Gaussians from landscape images with fluid content is useful but seems a bit too specific/narrow. If this paper can handle general 4D content, it would have a very large impact.
>
> **Response:** Eulerian flow is a concept from fluid dynamics commonly used in single-image animation methods. For example, Holynski et al. [1] and subsequent works have leveraged Eulerian flow to generate diverse looping videos. Recently, these methods have been extended beyond fluid motion to other domains, such as clouds [2] and clothing motion [3]. **In Appendix C**, we demonstrate that by replacing Holynski et al. [1] with another Eulerian flow-based model [2] in our framework, the 3D-MOM module still performs effectively, generating consistent 3D motion.
>
> Furthermore, our framework is designed to be broadly compatible with any single-image animation model capable of animating specified regions through masks. Handling highly complex motions, such as human or articulated movements, presents inherent challenges due to the current limitations of single-image animation techniques. While recent diffusion-based methods show promise in generating diverse and intricate motions, they struggle to maintain spatiotemporal consistency, particularly across multi-view settings. As a result, our work focuses on a fundamental aspect of dynamic scene reconstruction: representing natural phenomena, such as fluids, clouds, and dynamic textures, which are essential for applications in visual effects, virtual environments, and immersive media.
>
> As single-image animation techniques continue to evolve and expand to broader content domains, our proposed approach stands ready to enable consistent 3D motion estimation across diverse content types. The modular nature of our framework ensures its adaptability, facilitating the reconstruction of 4D scenes and broadening its applicability to increasingly complex motion domains in the future.
>
> **References:**
>
> [1] Aleksander Holynski, Brian L Curless, Steven M Seitz, and Richard Szeliski. *Animating pictures with Eulerian motion fields.* Proceedings of the IEEE/CVF Conference on Computer Vision and Pattern Recognition, 2021.
> [2] Choi, Jongwoo, et al. *StyleCineGAN: Landscape Cinemagraph Generation using a Pre-trained StyleGAN.* Proceedings of the IEEE/CVF Conference on Computer Vision and Pattern Recognition, 2024.
> [3] Bertiche, Hugo, et al. *Blowing in the wind: Cyclenet for human cinemagraphs from still images.* Proceedings of the IEEE/CVF Conference on Computer Vision and Pattern Recognition, 2023.

---

> ### Author Response · Authors · 2024-11-25
> **Comment 2/3 for Reviewer aadJ**
>
> **Weakness 2:** Lack of theoretical innovation for ICLR. To address the task of fluid motion, this paper uses constant speed and direction. As 3DGS is a particle-like model, the theoretical contribution should be greater if this paper can present or show different physical-based models for fluids.
>
> **Response:** Thank you for your insightful suggestion. To the best of our knowledge, directly estimating 3D motion in a particle-like manner, as described, remains an unexplored area. This paper addresses the challenge by introducing the 3D Motion Optimization Module (3D-MOM), which indirectly estimates 3D motion by leveraging well-established 2D motion estimation methods and lifting them into consistent 3D motion for particle-like models such as 3D-GS.
>
> This indirect approach allows the integration of existing physical-based 2D motion models, such as those described in [1] and [2]. In this work, we validated the effectiveness of our method using Eulerian flow with constant speed and direction. However, the 3D-MOM is designed to be compatible with other physical-based 2D motion models, further extending its applicability.
>
> While our current focus is on leveraging well-established 2D motion models, this work lays the foundation for future explorations into more advanced particle-like 3D motion estimation methods. The modular nature of our 3D-MOM facilitates integration with more sophisticated motion models, broadening its applicability to complex physical phenomena.
>
> **References:**
>
> [1] Chuang, Yung-Yu, et al. *Animating pictures with stochastic motion textures.* ACM SIGGRAPH 2005 Papers. 2005. 853-860.
> [2] Jhou, Wei-Cih, and Wen-Huang Cheng. *Animating still landscape photographs through cloud motion creation.* IEEE Transactions on Multimedia 18.1 (2015): 4-13.
>
> ---
>
> **Weakness 3:** Presentation of figures. Figure 2’s motion content is hard to see. Too many small arrows and small squares hurt the quality of the figure.
>
> **Response:** We apologize for the inconvenience. To improve the quality and clarity of the figure, we have simplified and refined Figure 2 to make the motion content more visible and easier to understand.
>
> ---
>
> **Question 1:** What is the max temporal length the proposed method can achieve with good quality? It has potential value if this paper can handle a long temporal range (with generated content).
>
> **Response:** Thank you for your question. We have added experiments in **Appendix F** and [**project page**](https://gramnard.github.io/ICLR_3D_MOM/#section11) that address the maximum temporal length achievable with our method. The results demonstrate that our framework can generate Dynamic Scene Videos with natural motion and high fidelity for up to 360 frames. While this represents the limit we tested, longer videos can also be generated, with the primary trade-off being increased training time.
>
> Compared to existing diffusion-based approaches, which typically generate 16 to 30 frames per inference [1], [2], [3], [4], our framework benefits from the efficiency of explicit 4D Gaussian representations. This allows for reduced computational cost and faster rendering, making it well-suited for producing extended sequences.
>
> **References:**
>
> [1] Zhang, Shiwei, et al. "I2vgen-xl: High-quality image-to-video synthesis via cascaded diffusion models." arXiv preprint arXiv:2311.04145 (2023).
> [2] Xing, Jinbo, et al. "Dynamicrafter: Animating open-domain images with video diffusion priors." European Conference on Computer Vision. Springer, Cham, 2025.
> [3] Shi, Xiaoyu, et al. "Motion-i2v: Consistent and controllable image-to-video generation with explicit motion modeling." ACM SIGGRAPH 2024 Conference Papers. 2024.
> [4] Yu, Wangbo, et al. "Viewcrafter: Taming video diffusion models for high-fidelity novel view synthesis." arXiv preprint arXiv:2409.02048 (2024).

---

> ### Author Response · Authors · 2024-11-25
> **Comment 3/3 for Reviewer aadJ**
>
> **Question 2:** Can authors provide more technical insights? This paper builds upon a lot of exciting techs (depth prediction model, 2D motion models). I would like to know if the authors selected these methods by experiments or random or with insights. If this paper can improve this part, it can benefit future work in this field.
>
> **Response:** Thank you for your thoughtful question. Our work introduces a comprehensive framework for reconstructing 4D scenes tailored to Dynamic Scene Video, focusing on integrating diverse research fields through our core module, 3D-MOM. As you correctly pointed out, the individual modules within our framework—depth prediction, 2D motion estimation, video generation, and 4D Gaussian optimization—are modular and interchangeable, offering flexibility for future advancements.
>
> The selection of these components was guided by systematic experimentation and domain-specific insights to achieve high-fidelity 4D scene reconstruction. Below, we summarize the key models tested and our final choices:
>
> 1. **Depth Prediction Models:**
>    - **Tested:** ZoeDepth [1] and DPT (Vision Transformers for Dense Prediction) [2].
>    - **Final Selection:** ZoeDepth was chosen for its robust zero-shot accuracy and strong generalization across diverse input scenarios, which aligns with the requirements of our framework.
>
> 2. **2D Motion Estimation Models:**
>    - **Tested:** Holynski et al. Animating Pictures with Eulerian Motion Fields [3] and Controllable Animation of Fluid Elements in Still Images [4].
>    - **Final Selection:** Holynski et al. was selected for its ability to generate consistent looping animations, particularly for natural phenomena like fluids and clouds.
>
> 3. **Video Generators:**
>    - **Tested:** Text2Cinemagraph [5], SLR-SFS [6], and StyleCineGAN [7].
>    - **Final Selection:** SLR-SFS was chosen for its balance between computational efficiency and temporal coherence, ensuring smooth integration with our framework.
>
> 4. **4D Gaussian Optimization Models:**
>    - **Tested:** 4D Gaussian Splatting (4D-GS) [8] and Deformable 3D GS [9].
>    - **Final Selection:** 4D Gaussian Splatting (4D-GS) was selected for its superior performance in efficiently modeling spatiotemporal dynamics while maintaining high fidelity.
>
> By combining these carefully selected components, our framework achieves modularity and adaptability. This ensures compatibility with future advancements in single-image animation, video generation, and 4D Gaussian techniques, laying a strong foundation for broader applications and future work. Details on the experimental process and selection criteria for each component have been added to **Appendix G** for further clarity.
>
> ---
>
> **References:**
>
> [1] Bhat, Shariq Farooq, et al. "Zoedepth: Zero-shot transfer by combining relative and metric depth." arXiv preprint arXiv:2302.12288 (2023).
> [2] Ranftl, René, Alexey Bochkovskiy, and Vladlen Koltun. "Vision transformers for dense prediction." Proceedings of the IEEE/CVF international conference on computer vision. 2021.
> [3] Holynski, Aleksander, et al. "Animating pictures with eulerian motion fields." Proceedings of the IEEE/CVF Conference on Computer Vision and Pattern Recognition. 2021.
> [4] Mahapatra, Aniruddha, and Kuldeep Kulkarni. "Controllable animation of fluid elements in still images." Proceedings of the IEEE/CVF Conference on Computer Vision and Pattern Recognition. 2022.
> [5] Mahapatra, Aniruddha, et al. "Text-guided synthesis of eulerian cinemagraphs." ACM Transactions on Graphics (TOG) 42.6 (2023): 1-13.
> [6] Fan, Siming, et al. "Simulating fluids in real-world still images." Proceedings of the IEEE/CVF International Conference on Computer Vision. 2023.
> [7] Choi, Jongwoo, et al. "StyleCineGAN: Landscape Cinemagraph Generation using a Pre-trained StyleGAN." Proceedings of the IEEE/CVF Conference on Computer Vision and Pattern Recognition. 2024.
> [8] Wu, Guanjun, et al. "4d gaussian splatting for real-time dynamic scene rendering." Proceedings of the IEEE/CVF Conference on Computer Vision and Pattern Recognition. 2024.
> [9] Yang, Ziyi, et al. "Deformable 3d gaussians for high-fidelity monocular dynamic scene reconstruction." Proceedings of the IEEE/CVF Conference on Computer Vision and Pattern Recognition. 2024.

---

> ### Author Response · Authors · 2024-12-01
>
> **Dear Reviewer aadJ,**
>
> Thank you again for your thoughtful feedback and for taking the time to review our previous responses. As the discussion period is nearing its conclusion, we wanted to follow up to ensure that we have fully addressed your concerns and to share some additional updates that we believe further strengthen our contributions.
>
> We understand your concern regarding the perceived specificity of focusing on fluid content. However, we would like to highlight that fluid-like dynamics, including phenomena such as fluids, clouds, and smoke, are foundational to a wide range of applications in visual effects, immersive media, and virtual environments. Recent works such as **StyleCineGAN** (CVPR 2024) and **Simulating Fluids in Real-World Still Images** (ICCV 2023 Oral) emphasize the importance and ongoing research interest in modeling these dynamics, underscoring their relevance and impact.
>
> In response to your suggestion to explore broader applications beyond fluid dynamics, we conducted additional experiments on dynamic phenomena such as clouds and smoke. These results demonstrate that our framework generalizes effectively to different types of “fluid-like dynamics” beyond liquids. While clouds and smoke share continuous motion and flowing characteristics with fluid content, they present unique challenges in terms of spatial and temporal consistency. The results validate the adaptability and robustness of our method across diverse content domains. We have updated our [project page](https://gramnard.github.io/ICLR_3D_MOM/) with these experiments, where videos are now available for your review.
>
> While our current experiments are scoped to these dynamic phenomena due to practical constraints, we acknowledge the importance of expanding the range of validation to include more complex and diverse 4D content. The modular design of our framework ensures this adaptability by allowing for the integration of various single-image animation models. This flexibility lays the groundwork for extending our approach to handle increasingly intricate and varied motion domains as these techniques advance. We view this as a foundational contribution that can evolve alongside the field and address even broader applications in future research.
>
> We deeply appreciate your constructive feedback, which has been instrumental in helping us refine and communicate our work more effectively. If there are any further suggestions or concerns, we would be delighted to address them. Your insights have been invaluable, and we hope the additional results and clarifications align with your expectations.
>
> Thank you again for your time and consideration, and we look forward to any further comments you might have.
>
> **Best regards.**

---

### Official Review · Reviewer_TeT9 · 2024-10-29

**Soundness:** 2
**Presentation:** 2
**Contribution:** 2
**Rating:** 3
**Confidence:** 4

**Summary:**

This paper presents a framework to form the 4D Gaussians representation for dynamic video generated from the single landscape image. To be more specific, the author builds the point cloud based on the single depth estimation. By reprojecting it to the multi view images, the static 3D Gaussians can be optimized. Then, the author proposed consistent 3D motions estimation module to lift the predicted 2D motions into the 3D space and further achieve the consistent motion in the 3D space with further optimization. Finally, the motion and 3D guassian can be used to optimize the 4D Gaussians as the final representation. The experiment conducted on the dataset from Animating Pictures with Eulerian Motion Fields demonstrating that the results surpasses several baseline methods.

**Strengths:**

1. The experiment shows the performance surpass the baselines on the public dataset.
2. The paper tries to solve the multi view consistency issue in the 3D generation by leveraging minimizing the reprojection error for the motion prediction. It is a good to attempt for this common challenging issue in this field.

**Weaknesses:**

1. The Sec. 3.1 for 3D Gaussians generation seems to just follow the previous work, Luciddreamer. Please correct me there is any additional novel effort for this part.
2. Reprojecting the point cloud from the single view image to different views always leads to the holes, distortion and some other artifacts. A common idea to incorporate the generative model to fulfill the missing information. But it seems the author does not include anything related to this point. Please include more discussion if it is not necessary.
3. Minimizing the reprojection error for the predicted motions seems to be feasible in Sec. 3.2, However, this optimization is still based on the 2D information and can hardly achieve the 3D consistency. For example, in Structure from Motions, such method is always used refine the estimated camera poses and the position for the point clouds. But it does not consider the inter relationship between each 3D positions, so I am not sure that the 3D consistency can be achieved with this method.
4. Please provide more implementation details of the 3D Motion Optimization Module. It seems to be vague now.
5. The Sec. 3.3 for 4D Gaussians generation seems to just follow the previous work. Please correct me there is any additional novel effort for this part.
6. The experimental results will be more solid and comprehensive if more dataset and baselines can be included. For example, the author mentions that simply using animation based method cannot achieve satisfying results, so similar baselines can be included.
7. The paper presentation can be more compact if the part of just following the previous work can be shortened while emphasizing the novel part.

**Questions:**

Please refer to the weakness.

---

> ### Author Response · Authors · 2024-11-25
> **Comment 1/3 for Reviewer TeT9**
>
> We thank the reviewer for the careful reading and thoughtful comments. Below, we address the reviewer’s questions and provide detailed explanations. The corresponding revisions in the manuscript are marked in blue for clarity. We hope that the responses below, along with the updated manuscript, address the reviewer’s concerns thoroughly. Due to character limits on OpenReview, our response has been divided into three parts for ease of readability. Thank you for your understanding.
>
> **Weakness 1:** The Sec. 3.1 for 3D Gaussians generation seems to just follow the previous work, Luciddreamer. Please correct me if there is any additional novel effort for this part.
>
> **Response:** Thank you for your observation. As you noted, the 3D Gaussian optimization method described in Sec. 3.1 builds upon the approach introduced in LucidDreamer [1], as well as other widely used methods in the 3D scene generation field [1, 2]. We adopted this approach to ensure compatibility between the 3D scene generation and dynamic scene video domains, allowing our framework to effectively bridge these fields in future applications.
>
> While Sec. 3.1 primarily focuses on adopting existing 3D Gaussian optimization techniques, our goal was to design the framework with cross-domain compatibility in mind. Specifically, traditional 3D scene generation methods are centered on spatial expansion through generative models, whereas dynamic scene video emphasizes temporal expansion. By integrating the strengths of both, our framework paves the way for the novel task of 4D scene generation.
>
> To demonstrate this potential, we highlight how our method facilitates the combination of these fields in the experiments detailed in **Appendix B**, with additional video results now updated on the [project page](https://gramnard.github.io/ICLR_3D_MOM/#section5). We believe this compatibility-focused design enables new possibilities for both 3D and dynamic scene research.
>
> Thank you again for your thoughtful feedback, which has allowed us to better articulate the strengths and contributions of our approach.
>
> **References:**
>
> [1] CHUNG, Jaeyoung, et al. *Luciddreamer: Domain-free generation of 3d gaussian splatting scenes.* arXiv preprint arXiv:2311.13384, 2023.
> [2] OUYANG, Hao, et al. *Text2immersion: Generative immersive scene with 3d gaussians.* arXiv preprint arXiv:2312.09242, 2023.
>
> ---
>
> **Weakness 2:** Reprojecting the point cloud from the single view image to different views always leads to the holes, distortion and some other artifacts. A common idea is to incorporate the generative model to fulfill the missing information. But it seems the author does not include anything related to this point. Please include more discussion if it is not necessary.
>
> **Response:** Thank you for your insightful comment. You are correct that in general, 3D scene generation often involves large scenes and diverse camera views, which can lead to significant holes and artifacts. Generative models are indeed a common approach to address these challenges. However, our work focuses specifically on reconstructing dynamic scene videos from a single image. In this task, the range of camera movements is intentionally limited, resulting in much smaller holes and fewer distortions compared to traditional 3D scene generation. To address the remaining holes, we employed a simple yet effective linear interpolation method, which avoids the computational overhead and potential artifacts that can arise from generative models.
>
> Our experimental results demonstrate that this approach enables higher fidelity and fewer artifacts in the reconstructed videos compared to state-of-the-art models in the dynamic scene video domain. We appreciate your suggestion, and we included a more detailed explanation in the revised manuscript to clarify our approach.

---

> ### Author Response · Authors · 2024-11-25
> **Comment 2/3 for Reviewer TeT9**
>
> **Weakness 3:** Minimizing the reprojection error for the predicted motions seems to be feasible in Sec. 3.2. However, this optimization is still based on the 2D information and can hardly achieve the 3D consistency. For example, in Structure from Motions, such methods are always used to refine the estimated camera poses and the position for the point clouds. But it does not consider the inter-relationship between each 3D position, so I am not sure that the 3D consistency can be achieved with this method. Please provide more implementation details of the 3D Motion Optimization Module. It seems to be vague now.
>
>
>
> **Response:** Apologies for any confusion regarding the proposed 3D Motion Optimization Module (3D-MOM). As you correctly pointed out, our module estimates 3D information based on 2D data, similar to Structure from Motion (SfM), and does not explicitly consider inter-relationships between individual 3D positions. However, the proposed 3D-MOM parametrically models 3D motion and updates it by comparing the reprojected 2D motion with the estimated 2D motion across multi-view settings. This ensures consistency in the updated 3D parametric motion. While it may not provide an optimal solution for estimating 3D motion, achieving 3D consistency was prioritized, as inconsistent motions lead to unnatural movements in reconstructed 4D scenes (see **Figure 4**).
>
> For implementation: 3D motion is initialized using point clouds from the input image at two timestamps. The first point cloud is fixed, while the second is treated as a learnable parameter. Their difference models the 3D motion, which is reprojected into 2D space using camera parameters. The L1 loss between reprojected and estimated 2D motion maps is calculated across multi-view settings.
>
> We have expanded **Section 3.2.1** and **Section 3.2.2** in the manuscript and updated **Figure 2** for clarity. Thank you for your feedback—it helped us improve the explanation.
>
> ---
>
> **Weakness 4:** The Sec. 3.3 for 4D Gaussians generation seems to just follow the previous work. Please correct me if there is any additional novel effort for this part.
>
> **Response:** We appreciate your insightful comment. As you correctly noted, the 4D-GS [1] algorithm builds on prior works. However, our framework is broadly compatible with various 4D Gaussian models [2, 3, 4], rather than being tied to one. This ensures our approach works effectively even when replacing the underlying 4D Gaussian model with Deformable 3D GS [2], as shown in **Figure 12** (see **Appendix D** and [Project Page](https://gramnard.github.io/ICLR_3D_MOM/#section9)).
>
> Our contributions in Sec. 3.3 focus on two aspects:
>
> 1. **3D Motion Initialization:** Without this step, as shown in **Figure 5** and the [project page](https://gramnard.github.io/ICLR_3D_MOM/#section8), conventional models like Deformable 3D GS struggle with repetitive patterns in fluid regions, resulting in artifacts. By incorporating 3D motion initialization, our framework improves robustness and maintains high-quality results even in large landscapes with substantial motion.
>
> 2. **Two-Stage Training Approach:** Unlike prior works, we introduce a two-stage training process separating spatial and temporal reconstruction. Stage 1 uses multi-view images to capture scene geometry, while Stage 2 refines motion and deformation using sampled animated videos. This design enables efficient training and high-quality spatiotemporal reconstruction without generating videos for all views.
>
> We revised **Section 3.3** to reduce emphasis on 4D-GS architecture and highlight compatibility with various 4D Gaussian models and our novel contributions. Details are in **Appendix D**. Thank you for your thoughtful feedback.
>
> **References:**
>
> [1] Wu, Guanjun, et al. *4d gaussian splatting for real-time dynamic scene rendering.* Proceedings of the IEEE/CVF Conference on Computer Vision and Pattern Recognition. 2024.
> [2] Yang, Ziyi, et al. *Deformable 3d gaussians for high-fidelity monocular dynamic scene reconstruction.* Proceedings of the IEEE/CVF Conference on Computer Vision and Pattern Recognition. 2024.
> [3] Huang, Yi-Hua, et al. *Sc-gs: Sparse-controlled gaussian splatting for editable dynamic scenes.* Proceedings of the IEEE/CVF Conference on Computer Vision and Pattern Recognition. 2024.
> [4] Li, Zhan, et al. *Spacetime gaussian feature splatting for real-time dynamic view synthesis.* Proceedings of the IEEE/CVF Conference on Computer Vision and Pattern Recognition. 2024.

---

> ### Author Response · Authors · 2024-11-25
> **Comment 3/3 for Reviewer TeT9**
>
> **Weakness 5:** The experimental results will be more solid and comprehensive if more datasets and baselines can be included. For example, the author mentions that simply using animation-based methods cannot achieve satisfying results, so similar baselines can be included.
>
> **Response:** Thank you for your comment. In this field, it is common to use the dataset from Holinsky et al., which we have employed for the quantitative and qualitative results in the main paper.
>
> To provide additional comparisons, we collected data of global landmarks from online sources and compared the performance of our method with baseline models. Additional results of these experiments are in **Appendix A**.
>
> Regarding baselines, the Dynamic Scene field offers limited models, such as 3D-Cinemagraphy [1] and Make-it-4D [2]. Recent approaches like Dynamicrafter [3] and Motion-I2V [4] rely on diffusion models for cinemagraphy generation. However, Dynamicrafter lacks view-specific control, and Motion-I2V struggles with precise adjustments, producing inconsistent results across trials. Comparative results with T2V models have been added to the **Qualitative Results.**
>
> Our framework, modeled using 4D Gaussian Splatting (4D GS), offers distinct advantages over diffusion-based methods. Unlike these approaches, our framework renders along arbitrary camera trajectories while maintaining superior spatiotemporal consistency. Performance comparisons are available on our [**project page**](https://gramnard.github.io/ICLR_3D_MOM/#section4).
>
> | **Method**           | **PSNR ↑** | **SSIM ↑** | **LPIPS ↓** | **PIQE ↓** | **Immersion (%)** | **Realism (%)** | **Structural Consistency (%)** | **Quality (%)** |
> |-----------------------|------------|------------|-------------|------------|--------------------|------------------|-------------------------------|------------------|
> | DynamiCrafter [3]     | 14.98      | 0.81       | 0.23        | 24.58      | -                  | -                | -                             | -                |
> | Motion-I2V [4]        | 14.38      | 0.80       | 0.31        | 8.40       | -                  | -                | -                             | -                |
> | 3D-Cinemagraphy [1]   | 17.30      | 0.83       | 0.17        | 8.93       | 31.87             | 31.87           | 28.75                        | 30.31           |
> | Make-It-4D [2]        | 16.98      | 0.81       | 0.20        | 8.30       | 11.87             | 10.31           | 8.43                         | 9.06            |
> | **Ours**              | **20.57** | **0.90**   | **0.14**    | **7.80**   | **56.25**         | **57.81**       | **62.81**                     | **60.25**       |
>
> **References:**
>
> [1] Li, Xingyi, et al. *3D cinemagraphy from a single image.* Proceedings of the IEEE/CVF Conference on Computer Vision and Pattern Recognition. 2023.
> [2] Shen, Liao, et al. *Make-it-4d: Synthesizing a consistent long-term dynamic scene video from a single image.* Proceedings of the 31st ACM International Conference on Multimedia. 2023.
> [3] Xing, Jinbo, et al. *Dynamicrafter: Animating open-domain images with video diffusion priors.* European Conference on Computer Vision. Springer, Cham, 2025.
> [4] Shi, Xiaoyu, et al. *Motion-I2v: Consistent and controllable image-to-video generation with explicit motion modeling.* ACM SIGGRAPH 2024 Conference Papers. 2024.
>
> ---
>
> **Weakness 6:** The paper presentation can be more compact if the part of just following the previous work can be shortened while emphasizing the novel part.
>
> **Response:** Thank you for your thoughtful feedback. We appreciate the opportunity to refine our manuscript and clarify the unique contributions of our work. We revised Sec. 3.1 and Sec. 3.3 to emphasize the novel aspects of our framework.
>
> In Sec. 3.1, we streamlined the discussion of 3D Gaussian optimization to focus on bridging 3D scene generation and dynamic scene video. While this section adopts established methods, our innovation lies in designing a framework that enables seamless integration of these domains, paving the way for 4D scene generation with improved spatiotemporal consistency.
>
> In Sec. 3.3, we reduced emphasis on the 4D Gaussian architecture and highlighted two key contributions:
> 1. **3D motion initialization**, which overcomes limitations of conventional 4D Gaussian models by resolving repetitive patterns and artifacts in fluid regions.
> 2. **A two-stage training approach**, which separates spatial and temporal reconstruction for efficient training and high-quality results.
>
> Furthermore, we highlighted our framework’s compatibility with various 4D Gaussian models. Additional explanations and results are on our [**project page**](https://gramnard.github.io/ICLR_3D_MOM/#section9).
>
> These revisions better showcase how our framework extends beyond existing works, leveraging and enhancing established techniques to address challenges in dynamic scene video reconstruction.

---

> > ### Comment · Reviewer_TeT9 · 2024-11-28
> >
> > Thank you for your detailed response, adding new context and making the presentation more logically fluent. However, my concern is not adequately addresses.
> > 1. I understand that when the range of camera movements is limited, only small holes and few distortions can be yielded. However, according to the setting in the paper, the camera trajectory consists of 105 cameras in total, which make the movement of the camera beyond limited. I do concern the negative impact introduced by the forward warpping from the input image.
> > 2. I understand that author optimize the 3D motions by projecting them to each 2D frames, comparing with the 2D motions from the pre-train model. However, this solution seems to be trivial. Since there is no multi-view consistency from the pre-train flow estimation model, the consistency of the optimized 3D motions is questionable.
> > 3. Even though the author emphasize the compatibility with existing method, the achievement from each model is still trivial. Such over-claiming may hurt the validity of this work.

---

> > > ### Author Response · Authors · 2024-11-29
> > > **Comment 1/2 for Reviewer TeT9**
> > >
> > > **Question 1:** I understand that when the range of camera movements is limited, only small holes and few distortions can be yielded. However, according to the setting in the paper, the camera trajectory consists of 105 cameras in total, which make the movement of the camera beyond limited. I do concern the negative impact introduced by the forward warpping from the input image.
> > >
> > >
> > > **Response:** We sincerely apologize for any confusion caused by the explanation in the paper. The reviewer is correct that large camera movements during forward warping from a single image can introduce significant holes, potentially affecting 3D reconstruction. However, our experiments were conducted under two distinct settings:
> > >
> > > - **30 Trajectories:**
> > >   These were used in the experiments without the generative model, where we specifically limited camera movements to ensure consistency and minimize artifacts. In this setting, the lifted point cloud was reprojected using carefully controlled camera parameters, avoiding significant holes in the generated multi-view images. Any minor holes that appeared internally were effectively addressed through interpolation, while outer regions did not impact the 3D reconstruction. We have included videos of the 30 trajectory on our [Project Page](https://gramnard.github.io/ICLR_3D_MOM/#section12) for your reference, where you can verify the absence of significant distortions.
> > >
> > > - **105 Trajectories:**
> > >   These were used in experiments involving the generative model. In this setting, the point cloud is progressively expanded and rendered using these trajectories. The generative model inherently handles the larger set of trajectories by filling in gaps during the progressive expansion process, preventing the occurrence of large holes.
> > >
> > > We sincerely regret the error in the paper and apologize for any confusion it may have caused. We will ensure this is corrected in future revisions to provide clearer and more accurate details. Thank you again for your thoughtful review and for giving us the opportunity to clarify.

---

> > > ### Author Response · Authors · 2024-11-29
> > > **Comment 2/2 for Reviewer TeT9**
> > >
> > > **Question 2:** I understand that author optimize the 3D motions by projecting them to each 2D frames, comparing with the 2D motions from the pre-train model. However, this solution seems to be trivial. Since there is no multi-view consistency from the pre-train flow estimation model, the consistency of the optimized 3D motions is questionable.
> > >
> > > **Response:** Thank you for your thoughtful feedback. We appreciate the opportunity to provide further details on our approach to consistent 3D motion estimation. While the paper outlines our method, we would like to clarify some critical aspects to better address your concerns.
> > >
> > > The key novelty of our approach lies in how we establish consistent starting and ending points for 3D motion, leveraging both 3D point clouds and 2D motion estimation. To achieve consistent 3D motion, it is critical to maintain alignment between the starting and ending points of motion across both 2D and 3D domains. Our method addresses this by:
> > >
> > > 1. **Establishing a Consistent Starting Point:**
> > >    Using a single image, we generate a 3D point cloud through depth estimation. This 3D point cloud serves as the starting point of motion in the 3D domain. When reprojected back into 2D views, the 3D starting points align with the corresponding pixel positions in 2D, ensuring consistency between the 2D and 3D domains.
> > >
> > > 2. **Estimating the Ending Point in 2D:**
> > >    In the 2D domain, we use a single-image animation model to estimate optical flow from the reprojected starting points. These motion vectors define the ending points of motion in 2D for each view.
> > >
> > > 3. **Optimizing for 3D Consistency:**
> > >    By minimizing reprojection error across all views, our optimization process refines the 3D motion ending points iteratively, ensuring that they align with the 2D motion estimations. This multi-view optimization ensures consistency across viewpoints and enables accurate reconstruction of the 3D motion trajectory.
> > >
> > > While minimizing reprojection error is widely used, it is not the central focus of our approach. Instead, the key contribution lies in our unified framework, which integrates 3D point clouds and 2D motion estimations to achieve consistent 3D motion reconstruction for dynamic scene video. The consistent alignment of starting and ending points across both domains is a critical contribution that enables robust 3D motion estimation from a single image. This approach is distinct because:
> > >
> > > - **Unified 2D-3D Consistency:**
> > >   Our method establishes consistency by integrating 3D point clouds (spatial information) with 2D motion estimations (temporal dynamics). This ensures that the reconstructed 3D trajectories are aligned not only within individual views but also across multiple viewpoints, bridging the gap between 2D and 3D representations effectively.
> > >
> > > - **Ensuring 3D Consistency:**
> > >   Our method ensures multi-view consistency by aligning predicted 2D motions across views through the 3D-MOM. Notably, the 2D motions inherently reflect spatial coherence within each view, as single-image animation models estimate flows that align with the spatial relationships of pixels. By aligning these spatially consistent 2D motions across views, we achieve consistent 3D motion representations without explicitly modeling inter-relationships between 3D positions.
> > >
> > > - **Dynamic Scene Applicability:**
> > >   Our framework adapts to dynamic scenes where explicit 3D motion estimation is not feasible due to the lack of pre-trained models. The integration of 2D and 3D motion ensures robust handling of such challenging scenarios.
> > >
> > > We hope that this explanation complements the content of our paper and provides clarity on the significance of our unified framework for consistent 3D motion estimation. Thank you for your valuable feedback, which has helped us articulate our contributions more effectively.
> > >
> > > ---
> > >
> > > **Question 3:** Even though the author emphasize the compatibility with existing method, the achievement from each model is still trivial. Such over-claiming may hurt the validity of this work.
> > >
> > > **Response:** Thank you for highlighting your concerns regarding the perceived over-claiming of our work. While compatibility with existing methods is an inherent feature of our framework, it serves as a practical benefit rather than the central focus. The primary contribution of our work lies in presenting a unified framework for consistent 3D motion estimation, as previously detailed.
> > >
> > > We greatly appreciate your feedback, which has allowed us to clarify the intent and scope of our contributions. Thank you for the opportunity to address this important point.

---

> > > > ### Author Response · Authors · 2024-12-01
> > > >
> > > > **Dear Reviewer TeT9,**
> > > >
> > > > Thank you again for your thoughtful feedback and for taking the time to review our responses. We wanted to follow up to ensure that we have adequately addressed your concerns, particularly regarding the perceived triviality of our approach for consistent 3D motion estimation.
> > > >
> > > > As mentioned earlier, to achieve consistent 3D motion, it is critical to maintain alignment between the starting and ending points of motion across both 2D and 3D domains. Our framework is designed to address this challenge through a unified pipeline that integrates 3D point clouds and 2D motion estimations. By leveraging this integration, we ensure robust multi-view consistency and spatial coherence, effectively bridging the gap between 2D and 3D domains.
> > > >
> > > > While our approach incorporates existing tools such as 2D motion estimation model and reprojection optimization, the novelty lies in how these elements are combined into a cohesive framework. This design not only addresses the challenges of consistent 3D motion but also extends compatibility to advanced applications, such as 4D scene generation, as demonstrated through the experimental results and videos shared on our [project page](https://gramnard.github.io/ICLR_3D_MOM/#sectionRebuttal).
> > > >
> > > > We believe this unified framework represents a meaningful contribution, particularly given the lack of pre-trained models for explicit 3D motion estimation in dynamic scenes. However, we value your perspective and would greatly appreciate any additional suggestions or questions you may have regarding the novelty or implementation of our method.
> > > >
> > > > Once again, we sincerely thank you for your invaluable feedback, which has been instrumental in helping us articulate and refine the contributions of our work. Your insights have greatly enriched our understanding, and we look forward to hearing any further thoughts you might have.
> > > >
> > > > **Best regards.**

---

> > > > > ### Author Response · Authors · 2024-12-03
> > > > > **Follow-up on Reviewer TeT9 Feedback - Request for Reconsideration**
> > > > >
> > > > > **Dear Reviewer TeT9,**
> > > > >
> > > > > Thank you once again for your thoughtful feedback and the opportunity to refine our work. Based on your concerns, we have made substantial improvements to highlight the novelty and robustness of our approach:
> > > > >
> > > > > 1. **Clarified Novelty of 3D Motion Optimization Module (3D-MOM):**
> > > > >    - Our method uniquely integrates 3D point clouds (spatial information) with 2D motion estimations (temporal dynamics) to achieve consistent 3D motion representation.
> > > > >    - By aligning spatially coherent 2D motions across views, we establish robust multi-view consistency without relying on explicit inter-relationship modeling between 3D positions.
> > > > >    - Our framework is particularly suited for dynamic scenes, where pre-trained 3D motion models are unavailable, ensuring reliable performance in challenging scenarios.
> > > > >
> > > > > 2. **Expanded Experimental Results:**
> > > > >    - Added comparisons on new datasets, such as global landmarks and "in-the-wild" scenes, demonstrating the versatility and robustness of our approach.
> > > > >    - Incorporated additional baselines (e.g., VividDream, ViewCrafter) and user study metrics on realism and immersion, which highlight the superior spatiotemporal consistency of our method (Table 1).
> > > > >
> > > > > 3. **Addressed Concerns on Forward Warping and Camera Trajectories:**
> > > > >    - Clearly differentiated the settings for 30 vs. 105 camera trajectories. Our experiments confirm minimal artifacts with the use of generative models, as detailed in the revised manuscript and additional results on our [project page](https://gramnard.github.io/ICLR_3D_MOM/#section12).
> > > > >
> > > > > 4. **Commitment to Community Impact:**
> > > > >    - We are fully committed to releasing the full code, pretrained models, and implementation details upon acceptance to ensure reproducibility and to contribute to advancing the field.
> > > > >
> > > > > We sincerely hope these efforts address your concerns about novelty, clarity, and robustness. We kindly request you to reconsider your score based on these comprehensive updates. Your thoughtful feedback has been instrumental in refining our work, and we remain open to any further suggestions or clarifications.
> > > > >
> > > > > Time is limited as the discussion period nears its end, and we truly hope our efforts and revisions meet your expectations.
> > > > > Thank you once again for your time, insights, and dedication.
> > > > >
> > > > > **Best regards,**

---

### Author Response · Authors · 2024-11-25
**Summary of Revision**

### **Summary of Revisions**

We thank the reviewers for their valuable feedback, which significantly improved our work. Below, we summarize the key updates. Additional results, including video demonstrations, are available on our [project page](https://gramnard.github.io/ICLR_3D_MOM/#sectionRebuttal).

---

### **Key Updates**

#### **Project Page Enhancements**
1. **4D Scene Generation**: Demonstrated the pipeline’s compatibility with existing 3D Scene Generation models and validated its performance using comparative experiments with VividDream and ViewCrafter (R3).
2. **Extended Results**: Included experiments on global landmarks and "in-the-wild" datasets, showcasing robustness across diverse scenarios (R1).
3. **Long Video Generation**: Addressed computational challenges with diffusion-based models, achieving up to 360 frames with consistent quality (R2).
4. **User Study Results**: Added subjective evaluations for realism, immersion, and quality, highlighting our approach’s superiority over baselines (R4).

---

#### **Main Paper Revisions**
1. **Clarified Methodology**: Updated **Figure 2** for better visualization of the 3D-MOM pipeline, ensuring spatiotemporal consistency by lifting 2D motion into 3D space. Expanded explanations in **Sec. 3.2** and **Sec. 3.3** (R1, R2, R4).
2. **Application to 4D**: Added **Sec. 4.5** to demonstrate 4D scene generation and modular compatibility (R3, R4).

---

#### **Appendix Updates**
1. **Comparative Analysis**: Included results comparing our method with baseline models (VividDream, ViewCrafter, 3D-Cinemagraphy) in **Appendix B** (R3).
2. **Motion Diversity**: Demonstrated compatibility with StyleCineGAN for handling diverse motions in **Appendix C** (R2).
3. **Extended Temporal Range**: Detailed experiments on maximum achievable temporal length in **Appendix F** (R2).
4. **Module Selection Insights**: Summarized experimental choices for key components in **Appendix G** (R2).

---

### **Key Contributions**
1. **3D Motion Optimization Module (3D-MOM):** Ensures spatiotemporal coherence by bridging 2D motion estimation with consistent 3D motion representation.
2. **Practical Applications:** Showcased compatibility with diverse scenarios, extending beyond fluid dynamics to general 4D scene generation.

We believe these revisions address the reviewers’ concerns and strengthen the paper’s contributions. Thank you for your thoughtful feedback!

---

### Author Response · Authors · 2024-12-04

**Dear Reviewers and Area Chair,**

We deeply appreciate the thoughtful feedback and rigorous discussions throughout the review process. Your insights have been instrumental in refining our work, and we are truly grateful for your time and effort in evaluating our submission. As we conclude the discussion phase, we would like to summarize the key contributions of our work and the substantial improvements made in response to your comments.

### **Key Contributions**

1. **Framework for Consistent 3D Motion Estimation:**
   - Our 3D Motion Optimization Module (3D-MOM) bridges 2D motion estimation with 3D representation to achieve multi-view consistency without relying on explicit inter-relationship modeling between 3D positions.
   - By focusing on dynamic scenes where pre-trained 3D motion models are unavailable, our method addresses a significant gap in the field and provides a scalable solution for real-world applications.

2. **Extensive Experimental Validation:**
   - We have expanded our experiments to include additional datasets (e.g., global landmarks, "in-the-wild" scenes) and added new baselines (e.g., DynamiCrafter, Motion-I2V) for comparison. These results demonstrate the versatility and robustness of our approach across diverse scenarios.
   - A user study evaluating realism, immersion, and structural consistency further highlights the superiority of our method over state-of-the-art approaches, as detailed in Table 1 and our project page.

3. **Adaptability and Broader Potential:**
   - While our work currently focuses on fluid-like dynamics, additional experiments on clouds and smoke demonstrate its adaptability to a broader range of phenomena. This modular design ensures compatibility with advancements in single-image animation, making it a foundational framework for future research.
   - We also appreciated Reviewer NyCT's recognition of the broader applicability of our framework to 4D scene generation. The experiments on dynamic phenomena further validate the robustness and versatility of our approach, addressing the need for adaptability in broader applications.

4. **Commitment to Reproducibility:**
   - Upon acceptance, we will release the full implementation, pretrained models, and detailed documentation to ensure reproducibility and facilitate further exploration by the research community.

### **Final Remarks**

This work represents our commitment to advancing the field of single-image-based dynamic scene reconstruction by addressing critical challenges in motion consistency and spatiotemporal representation. Through your feedback, we have significantly strengthened the clarity, robustness, and scope of our approach. We believe this work lays the groundwork for broader applications in immersive media, visual effects, and virtual environments.

We trust that the revisions and additional results provided during the discussion phase demonstrate our dedication to addressing all concerns raised by the reviewers. Your collective feedback has greatly enriched our work, and we hope for a positive outcome.

Thank you once again for your valuable feedback and for considering our submission.

**Best regards,**
The Authors

---

### Meta-Review · Area_Chair_KneR · 2024-12-21

**Metareview:**

This paper introduces a framework to optimize 4D Gaussian Splatting representations from a single landscape image. It addresses the challenge of multiview consistency by minimizing reprojection error for motion prediction and proposes the 3D-MOM method to ensure smooth animations. Experimental results demonstrate that the framework outperforms baseline methods on a public dataset.

While initial concerns were raised about the lack of sufficient baselines and comparisons, these were addressed in the rebuttal with additional evaluations. A major concern is about the novelty of the approach. Although the individual components are built on prior work, the paper effectively combines them into a cohesive framework for the innovative application of 4D landscape scene modeling. Moreover, the framework emphasizes compatibility with existing models, offering flexibility for integrating newly developed methods to improve results. This was exemplified by its successful integration with existing 3D scene generation models to synthesize novel content during the rebuttal period.

**Additional Comments On Reviewer Discussion:**

Initially, the majority of reviewers were negative about the paper and raised several concerns.

One issue was the novelty of 3D Gaussian generation. While the work builds on a prior approach to 3D Gaussian optimization, the rebuttal clarified that it applies this method to the novel task of 4D scene generation. Reviewers also questioned the originality of the consistent 3D motion estimation, to which the rebuttal responded by emphasizing the contribution of the cohesive framework introduced in the paper.

Reviewers further recommended including more datasets and baselines. In response, the revision incorporated comparisons with DynamiCrafter and Motion-I2V, and at the reviewers' request, a comparison with VividDream was also added. Following reviewer suggestions, the rebuttal also demonstrated integrating the proposed method with existing 3D scene generation models to synthesize novel content, showcasing the framework's flexibility. A user study was also included to enhance the evaluation.

Another concern was the handling of holes in reprojected point clouds. The rebuttal explained that small holes caused by minor camera motions can typically be resolved using simple linear interpolation, while larger holes are addressed effectively with generative models.

In the end, the majority of reviewers were satisfied with the rebuttal and became optimistic about the paper.

---

### Decision · Program_Chairs · 2025-01-22

Accept (Poster)